# An allosteric inhibitor of RhoGAP class-IX myosins suppresses the metastatic features of cancer cells

Despoina Kyriazi [1], Lea Voth[1], Almke Bader [1], Wiebke Ewert [1,2], Juliane Gerlach[3], Kerstin Elfrink[4], Peter Franz [1], Mariana I. Tsap[5], Bastian Schirmer [6], Julia Damiano-Guercio[1], Falk K. Hartmann[1], Masina Plenge[7], Azam Salari[8], Dennis Schöttelndreier[1], Katharina Strienke [1], Nadine Bresch[1], Claudio Salinas[1], Herwig O. Gutzeit[9], Nora Schaumann[10], Kais Hussein[11], Heike Bähre[12], Inga Brüsch[13], Peter Claus [14], Detlef Neumann[6], Manuel H. Taft [1], Halyna R. Shcherbata [5], Anaclet Ngezahayo [7], Martin Bähler [4], Mahdi Amiri [8], Hans-Joachim Knölker [3], Matthias Preller [1,2] & Georgios Tsiavaliaris[1] ✉

Aberrant Ras homologous (Rho) GTPase signalling is a major driver of cancer metastasis, and GTPase-activating proteins (GAPs), the negative regulators of RhoGTPases, are considered promising targets for suppressing metastasis, yet drug discovery efforts have remained elusive. Here, we report the identification and characterization of adhibin, a synthetic allosteric inhibitor of RhoGAP class-IX myosins that abrogates ATPase and motor function, suppressing RhoGTPase-mediated modes of cancer cell metastasis. In human and murine adenocarcinoma and melanoma cell models, including three-dimensional spheroid cultures, we reveal anti-migratory and anti-adhesive properties of adhibin that originate from local disturbances in RhoA/ROCK-regulated signalling, affecting actin-dynamics and actomyosin-based cell-contractility. Adhibin blocks membrane protrusion formation, disturbs remodelling of cell-matrix adhesions, affects contractile ring formation, and disrupts epithelial junction stability; processes severely impairing single/collective cell migration and cytokinesis. Combined with the non-toxic, non-pathological signatures of adhibin validated in organoids, mouse and *Drosophila* models, this mechanism of action provides the basis for developing anti-metastatic cancer therapies.

Metastasis is a genetically driven, multifactorial process of tumour dissemination, and the main cause of cancer therapy failure and mortality[1]. Tumour cells disseminate by migration, either collectively as sheets and clusters[2], or individually, where single cells transition through a mesenchymal- and/or amoeboid type of migration[3] to escape from the primary tumour and invade target organs to establish new connective attachments, followed by unrestrained growth and proliferation[4]. Single and collective cell migration, both share common pathways of receptor-mediated stimulation[5,6] that are tightly regulated via signalling cascades involving members of the Ras homologous (Rho) family of small guanosine triphosphatases (GTPases), including Rho, Rac, and Cdc42[7–9]. By cycling between GDP-bound (inactive) and GTP-bound (active) states, RhoGTPases control actin dynamics and actomyosin-based contractility, as well as the remodelling of cell–cell and cell–matrix adhesions, which enables cells to reorganize the cytoskeleton during migration, promoting further the establishment

---

of cellular connections and surface-adherences[9,10]. Aberrant RhoGT-Pase signalling is considered a dominant driving force of metastasis and cancer progression[11]. Particularly, oncogenic mutations in RhoGTPases and their regulators[12], excessive receptor signalling, and altered effector activity patterns, are factors that stimulate cells to gain pro-migratory capabilities and acquire highly invasive, proliferative phenotypes that promote dissemination and metastasis formation[13]. Thus, targeted interference of Rho-associated signalling cues has become a viable and increasingly investigated strategy for suppressing cancer metastasis[14].

Drug discovery efforts have primarily focused on targeting oncogenic events responsible for the activation of signalling cascades, such as guanine nucleotide-exchange factors (GEFs)[15], which act upstream and accelerate GDP to GTP exchange rendering the RhoGTPase active. The inhibition of down-stream effectors, such as Rho-associated protein kinases (ROCK1 and ROCK2) and the myotonic dystrophy kinase-related Cdc42-binding kinase (MRCK), has emerged as a complementary strategy to suppress the metastatic features of cancer cells[16–18]; however, the therapeutic potential of most inhibitors has remained limited[19]. Lack of target selectivity, side effects, and development of resistances have yet prevented positive responses to treatments and therapeutic breakthroughs[20].

A promising, yet elusively explored approach to target the metastatic properties of cancer cells, particularly those related to enhanced migration and invasiveness, is to gain control over the activity of GTPase-activating proteins (GAPs), the negative regulators of RhoGTPases, which act as key signal transducers of processes controlling cytoskeletal dynamics, cell polarity, cell migration, cell-cycle progression, and cytokinesis. Besides a few proto-oncogenic features[21,22], the majority of the GAPs can be classified as tumour-suppressors[15,23]. They downregulate RhoGTPase effector functions through stimulation of the GTP hydrolysis reaction, converting the GTPase to an inactive off-state[24]. To fine-tune the process of activation and termination of RhoGTPase signalling that governs cytoskeleton dynamics and controls actomyosin-based cell contractility, RhoGAPs obey tight spatiotemporal regulation. This involves distinct mechanisms of hormonal modulation including cytokines and growth factors, as well as mechanisms of mechanotransduction[25]. Thus, drugs that are capable of enhancing and/or locally controlling RhoGAP activity provide a means to suppress the adhesive and migratory properties of cancer cells, and thus metastasis, which according to current literature has yet not been achieved[20].

Among the various RhoGAPs found in mammals[26], the two class-IX myosin isoforms myosin-9a (Myo9a) and myosin-9b (Myo9b) are particularly interesting as anti-metastatic targets. As motorized Rho-GAPs, they utilise the in-built motor function to move processively along actin filaments, which enables self-transportation[27] and allows for spatial and temporal control over Rho-signalling[28]. Additionally, class-IX myosins actively participate in the formation and organization of focal and cohesive cell structures by acting as motorized scaffolding proteins with actin cross-linking capacities at the membrane, contributing to cell polarization, focal adhesion stability, and junctional integrity of cell–cell connections[28–31]. Myo9a has been implicated in controlling tight and adherence junctions during multicellular self-assembly and collective cell migration[31–34], whereas the Myo9b isoform is required for membrane protrusion formation and focal adhesion dynamics regulating the protrusive and contractile forces during random and directional cell movements[28,29,35].

The relevance of class-IX myosins as druggable targets is emphasised by their involvement in diseases including inflammatory bowel and coeliac diseases[36], pancreatitis[37], myasthenic syndromes[38], and cancer[36,39–41]. Particularly, Myo9b activity has been shown to interfere with the tumour-suppressive Slit/Robo signalling pathway[42], which is deregulated in many cancer types and promotes metastasis with poor prognosis[42–45].

In view of the critical involvement of class-IX myosins in RhoA signalling and processes of cell polarization, migration, adhesion, and cell–cell contact formation, this family of motorized GAPs appears to be a promising target for a drug-based intervention to treat metastatic cancer. A small molecule drug that specifically inactivates the motor function of class-IX myosins, should thus provide a means to disturb the molecular interplay of signalling events that promote cancer cells to acquire a migratory and invasive phenotype, and consequently act as an antimetastatic agent[46].

Here, we identified and characterized a potent and selective myosin class-IX inhibitor that acts as a RhoGTPase modulator and effectively suppresses metastatic features of cancer cells[19].

## Results

### Adhibin is a selective and allosteric inhibitor of class-IX myosins

We set out to identify a myosin class-IX specific inhibitor by screening an in-house compound library comprising halogenated pseudilins and carbazole derivatives, developed to target the ATPase activity of myosins[47,48]. Mechanism-based functional screening with recombinant and native myosin constructs from different organisms and classes led to the identification of four potent Myo9 inhibitors following the synthetic route as illustrated (Fig. 1A, B), with one particular derivative, 3,6-dibromo-1-(hydroxymethyl)carbazole (12), called adhibin due to its inhibitory properties on cell adhesion, as the most selective hit compound. Adhibin effectively inhibited basal and actin-activated ATPase activity of mammalian and invertebrate Myo9s in the low micromolar range with $IC_{50}$ values of $2.5 \pm 0.2\,\mu M$ and $2.6 \pm 0.2\,\mu M$ (Fig. 1C and Supplementary Table 1), respectively, but not skeletal muscle myosin (skHMM) (Fig. 1D), β-cardiac muscle myosin-2 (β-cardMyo2), and unconventional class-V myosins (Fig. 1E), or the three human non-muscle myosin-2 isoforms (NM2A,−2B,−2C) (Fig. 1F). Low micromolar concentrations inhibited Myo9 motor function substantially, reducing sliding velocity and motile fraction of actin filaments in a reproducible and reversible manner (Fig. 1G, H). Importantly, the motor properties of other tested myosins remained unaffected (Fig. 1G).

To gain insights into the binding mode of the carbazoles and elucidate the structural basis of inhibition, we conducted a series of crystallization trials with various myosin constructs and the hit compounds, finally succeeding to co-crystalize the motor domain of the amoeboid class-II model myosin DdMyo2 in complex with the structural analogue of adhibin (compound 5), namely 3,4-dibromo-6-methyl-9H-carbazol-1-ol, which displays comparable, but less selective properties of inhibition, additionally recognizing the amoeboid class-II myosin (DdMyo2) and class-V myosin (DdMyo5a) as targets (Fig. 1I). The 3.2 Å X-ray structure (PDB: 6Z2S, for diffraction data statistics and structure refinement see Supplementary Table 2) shows the myosin with bound 5 in the pre-power stroke conformation (r.m.s.d. = 0.434 Å)[49] and reveals the inhibitor binding site in close proximity to the N-terminal part of the central relay helix in the inner motor domain core, near the large cleft formed by the upper and lower 50 K domains, approximately 13 Å apart from the nucleotide-binding site (Fig. 1J). Structural elements responsible for communicating conformational changes along the myosin ATPase cycle, such as the relay helix, the strut loop, and the W-helix, which form a central hub in the myosin motor domain for effective force production[50], flank the inhibitor binding site (Fig. 1K). The site buries a total protein surface area of 302.9 Å². This pocket is not visible in ligand-free DdMyo2 pre-power stroke structures and only formed by rearrangements of the side chains of residues S266 (β7-loop), E467 (relay helix), and K587 (strut) upon binding of the carbazole (Supplementary Fig. 1A). The aromatic carbazole moiety interacts primarily through hydrophobic contacts with F466 (relay helix) and V630 (W-helix), as well as cation-π interactions with K423 (O-helix) (Supplementary Fig. 1B). Four hydrogen bonds stabilize the inhibitor in its binding position: S465 (relay helix) and K587 (strut loop) interact with the hydroxy group of the ligand,

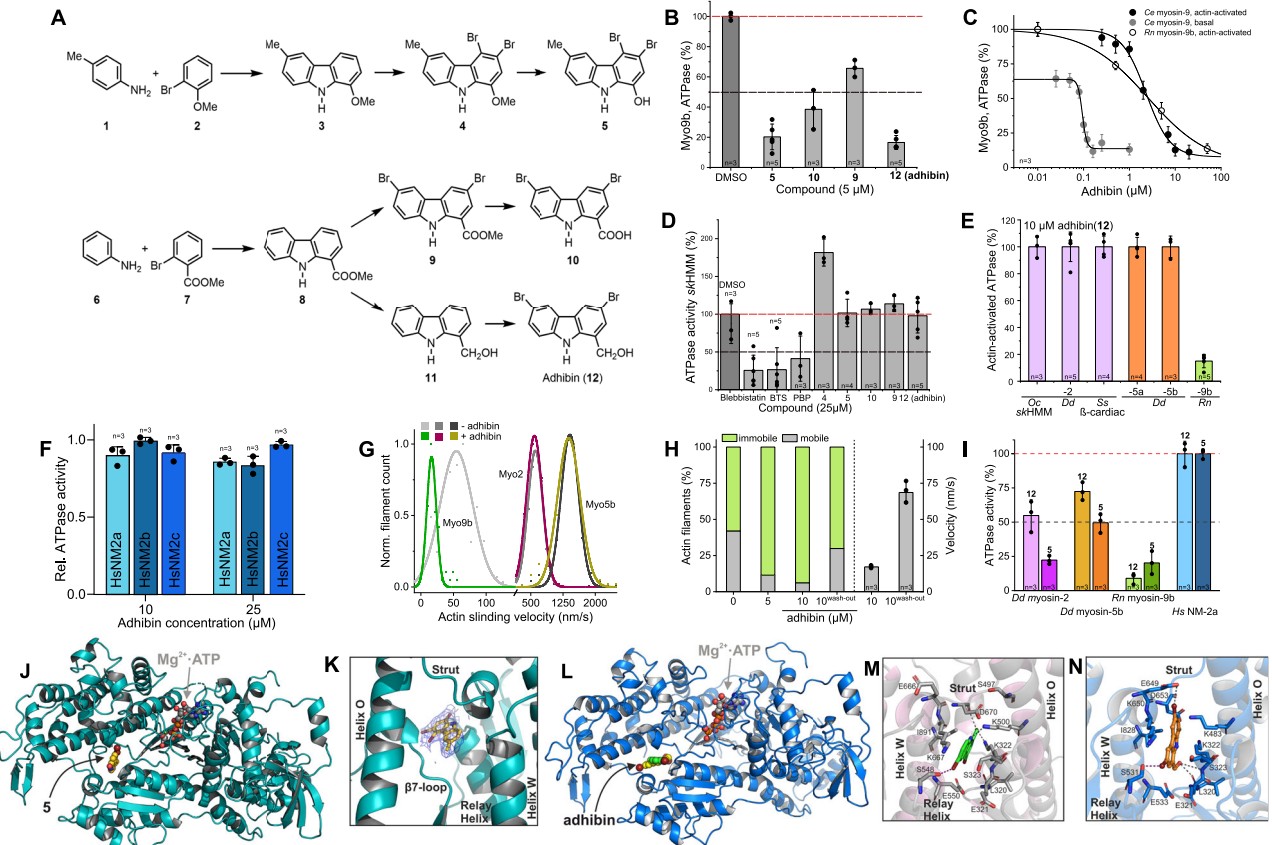

**Fig. 1 | Identification of adhibin as a myosin class-IX specific inhibitor.**
**A** Chemical structures and synthetic routes of compounds (details provided in
'Materials and Methods'). **B** Efficacy of compounds to inhibit *Rn*myosin-9b actin-
activated ATPase activity. **C** Inhibitory effect of adhibin on *Ce*myosin-9 and
*Rn*myosin-9b basal and actin-activated ATPase activities. **D** Effect of compounds on
basal ATPase activity of skeletal myosin-2 (*sk*HMM) in comparison to blebbistatin,
BTS, and PBP. **E** Effect of adhibin (**12**) on actin-activated ATPase of myosins from
class-II, -V and -IX. **F** Inhibitory effect of adhibin on actin-activated ATPase activity of
NM2A, 2B, and 2C. **G** Effect of adhibin on the motile activity of *Ce*myosin-9,
*Dd*myosin-2 and *Dd*myosin-5b. **H** Motile vs immotile actin filaments after incuba-
tion with adhibin. Recovery of motile properties after wash-out of adhibin. Data

derived from measurements of 80–200 filaments. **I** Inhibition of actin-activated
ATPase activity of different myosins by **5** and **12** at 25 μM concentrations. Data are
represented as mean ± S.D. **J** Crystal structure of *Dd*myosin-2 motor domain in
complex with ADP-VO₃ and adhibin analogue **5**. **K** The $2F_o - F_c$ density map of **5**,
contoured at 1.0 σ. **5** binds to a pocket surrounded by the strut, β7-loop, relay helix,
O-helix and W-helix. **L** Homology model of Myo9a from rat showing the adhibin
binding site. **M** Close-up view of the adhibin binding pocket in Myo9a and (**N**) in the
homology model of rat Myo9b. Hydrogen (grey dashed lines), halogen bonds
(purple dashed lines). *n* is the number of experiments. All data are represented as
mean ± S.D.

and S465 (relay helix) and the main chain of E487 (relay helix) bind to
the nitrogen of the carbazole framework. The two bromide sub-
stituents are involved in halogen bonds with residues E264 and S266
(both in the β7-loop). The binding site is partially overlapping with the
binding pocket in a crop pathogen *Fusarium* class-I myosin that
accommodates the fungicide phenamacril[51] (Supplementary Fig. 1C).

To examine the binding mode of adhibin in our target protein, we
generated homology models of Myo9a and Myo9b, and used a blind
docking approach to find the preferred binding site, followed by tar-
geted docking for higher precision (Fig. 1L and Supplementary Data 1).
While redocking of the adhibin analogue **5** into the *Dd*Myo2 crystal
structure underpins the compound's binding position in the allosteric
pocket (Supplementary Fig. 1D), docking of adhibin into the Myo9
structural models identifies a slightly shifted binding position of
adhibin towards residues E649 and Y654 of the actin-binding strut
loop, placing the inhibitor in bonding distance to residue S480 of
helix-O (Supplementary Fig. 1E). In both model structures, adhibin
occupies the same binding site as **5** (Fig. 1L). The adhibin binding
pocket is next to that of the myosin-2 inhibitor blebbistatin[52,53] and
flanked by the strut loop and the helix-W, which translate nucleotide
binding and hydrolysis to cleft closure, important to establish strong-
actin binding for effective force production[50,54]. The binding of adhibin

in close proximity to these residues could impair the fractional occu-
pancies of weak-to-strong actomyosin transitions and thus motor
function[50,55]. Hydrogen bonds, hydrophobic interactions, and halogen
bondings stabilize the binding and orientation of adhibin in the Myo9
binding pocket, which is slightly different between the Myo9 isoforms
(Fig. 1M, N), but the coordinating residues corresponding to K322,
K500, S548, D670, and L320, K322, K483, S531, and E649, respectively,
are conserved, also among myosins from different classes (Supple-
mentary Fig. 1F). However, V630 (W-helix) in *Dd*Myo2 is replaced by
the bulkier I891 and I828 in *Rn*Myo9a and *Rn*Myo9b. This seems to
produce a slightly smaller pocket with a higher geometrical com-
plementarity to achieve stronger binding of adhibin. Other myosins
from different classes have smaller amino acids at this position, such as
Ala or Val, (Supplementary Fig. 1F). Given the known conformational
changes in this region along the actomyosin cycle, our analyses argue
that the adhibin binding pocket is transient, as reported for other
myosin inhibitors[56]. Transiently established interactions appear to play
a major role in positioning adhibin for stronger binding to Myo9s. The
importance of transition states through which allosteric effector
molecules modulate myosin activity has been recently demonstrated
for two cardiac myosin modulators, which target the same myosin
pocket despite adverse effects[57]. Taken together, the biochemical and

structural data imply that adhibin disturbs nucleotide state-dependent myosin cleft movements essential for efficient catalytic cycling and effective force transmission. Inhibition of ATP turnover and motor function by adhibin support a mechanism by which the drug prolongs the states of strong actin interactions, causing deceleration of product release and increase of the duty ratio[50], parameters essentially determining the motile properties of a myosin. Adhibin thus fulfils the requirements of a specific and potent inhibitor of myosin class-IX motor function.

## Adhibin suppresses cancer cell migration and adhesion

To investigate the effect of Myo9 motor inhibition in cancer cells, we chose different epithelial human and mouse-derived melanoma and carcinoma cell lines (A549, MLE-12, B16-F1, Caco-2, and Calu-3), which express both Myo9a and Myo9b encoding transcripts at comparable levels as healthy lung tissue or higher (Supplementary Fig. 2A, B). Both inhibitors displayed no cell toxicity in the concentration range up to $25\,\mu M$ (Supplementary Fig. 2C, D). Higher doses caused a hyperbolic decline in cell viability with the corresponding $EC_{50}$ that was higher than the $IC_{50}$ values (Supplementary Tables 1 and 3). Compared to the previously reported class-V and class-I myosin inhibitors PBP[48] and PClP[58,59], which are based on the pseudilin scaffold, or the class-II specific inhibitor blebbistatin with its analogues, which are frequently employed in cellular studies addressing mechanisms of trafficking, transport and cell contractility[60–64], adhibin displays the lowest cytotoxicity by up to two orders of magnitude, depending on the cell type. In A549 and MLE-12 cells, total Myo9a or Myo9b protein levels remained almost unaffected by overnight treatments with adhibin (Supplementary Fig. 2E–H). Long-term administration delayed cell growth (Supplementary Fig. 2I, J) and prolonged cell cycle progression as revealed by proliferation analysis with replication markers (Supplementary Fig. 2K, L). A prominent observation was that adhibin caused the separation of single cells and entire cell groups from confluently grown layers, forcing their complete detachment from the underlying matrix (Supplementary Fig. 2M and Supplementary Movie 1).

Prominent morphological changes of the cells included a gain in both, height and circularity, as well as reduced cell area (Fig. 2A–H). These size and shape changes hint at potential defects in the contractile and adhesive properties of the cells responsible for a) maintaining membrane tension and b) establishing stable connections with the underlying matrix and neighbouring cells. Cell scattering and compromised migration ability of the scratched monolayers (Supplementary Fig. 3A–D) indicate that adhibin interferes with cell–cell adherence and migration. This phenotype resembles that previously reported for Myo9a and Myo9b depleted cells, which similarly failed to collectively migrate and maintain their cell–cell contacts[29,31,34]. Cell spreading assays confirm the interfering effect on cell-matrix adhesion. Adhibin-treated cells increasingly failed to attach to the surface and spread after seeding as reflected in the high circularity parameter, smaller area, and absence of flattening, which were maintained for hours after drug treatment (Supplementary Fig. 3E–K). Importantly, non-adherent, floating cells were still viable (Supplementary Fig. 3L), consistent with the absence of any toxic effects at the concentration used. To test for potential defects in single-cell migration, a process that relies on highly dynamic focal adhesions and the formation of lamellipodial structures at the leading edge generating front-rear polarity, we used the melanoma B16-F1 cell line as a highly motile epithelial cancer cell model[65] expressing both Myo9 isoforms at comparable levels (Supplementary Fig. 4A, B). This cell line reacted with defects at considerably lower drug amounts than the adenocarcinoma cells, revealing polarity failures and whole cell rounding (Supplementary Fig. 4C) without affecting viability (Supplementary Fig. 4D). Like A549 and MLE-12 cells, B16-F1 cells similarly failed to establish stable matrix adhesions (Fig. 2I, J). Upon settling, cells

maintained their initial rounded shape and size, even after hours of drug administration (Supplementary Fig. 4E–G and Supplementary Movie 2). Notably, the addition of adhibin to already surface-adhered cells caused rounding and polarity loss, which was accompanied by a shrinkage in cell size (Supplementary Fig. 4H, I). Migration speed and mean square displacement declined in a dose-dependent manner (Fig. 2K–M and Supplementary Movie 3), while randomness of migration remained unaffected (Fig. 2N). Notably, even submicromolar adhibin concentration led to a reduction of migration by almost 2-fold (Fig. 2L). The defects can be compared with those of Myo9b[-/-] cells, which similarly fail to spread, establish a polarized shape, and migrate[29]. B16-F1 cells siRNA-silenced in *MYO9B* expression, reducing the Myo9b levels by approx. 80% (Supplementary Fig. 5A) did not respond to the inhibitor with the same pronounced changes in shape and migration as control cells (Supplementary Fig. 5B, C). Single-cell migration was slightly reduced (Supplementary Fig. 5D), indicating that adhibin is more potent in interfering with migration than the single silencing of one of the target genes, assuming that one Myosin-9 isoform could substitute the other.

Cell-cycle analysis by flow-cytometry and non-isotopic immunostaining revealed that long-term treatments with adhibin increase the detectable population of cells in the G1-phase, also affecting S-phase progression (Supplementary Fig. 4J, K). Since RhoGTPases and particularly RhoA are involved in regulating G1/S-phase transition by repressing cyclin-dependent kinase inhibitors (CDKI) and activating cyclin expression[20], the data indicate that adhibin interferes with the RhoGTPase signalling pathways through a Myo9 dependent mechanism. For comparison, macrophages isolated from the bone marrow of healthy mice did not show any obvious changes in morphology or adhesion upon adhibin treatment, even after long-term administration (Supplementary Fig. 4L). This could allow the tentative assumption that non-cancerous cells may be more tolerant to adhibin. To dissect the mechanism by which Myo9 inhibition causes cells to lose the adhesive and migratory properties, we studied down-stream effectors of the RhoA GTPase signalling cascade, focusing our investigations on proteins regulating cytoskeleton dynamics and cell contractility, including cell division.

## Adhibin interferes with Rho-mediated actin dynamics, actomyosin-based contractility, and cytokinesis

The migratory properties of cells are critically determined by actin remodelling and actomyosin-based contractility, both of which are controlled by the interconnected actions of the three major RhoGTPase signalling pathways RhoA/ROCK, Rac1/WAVE, and Cdc42/WASP[66]. First, we used confocal imaging to explore potential changes in actin architecture and dynamics, including studies of membrane protrusion formation and establishment of stable and functional cell-contractile actomyosin structures. These two highly dynamic processes are primarily governed by actin polymerization in the cell front and actomyosin-based force generation at the cell rear[66,67]. With $5\,\mu M$ adhibin, B16-F1 cells lost their typical polarized shape, developed multiple protrusive fronts (Fig. 3A, B), and displayed considerably thinner lamellipodia than control cells (Fig. 3C) visible within a time window of 6 h after drug administration. Higher adhibin concentrations induced further cell shrinkage and led to the complete disappearance of lamellipodia (Fig. 3A). Actin microspikes retained their typical length (Fig. 3D) but lacked membrane embedment, extended as finger-like protrusions beyond the cortical cell edges, and displayed reduced dynamics (Supplementary Movie 4). The protruding nature of microspikes in adhibin-treated cells is further highlighted in the increased ratio between microspike length and lamellipodium width (Fig. 3E). Additionally, lamellipodia contained significantly less F-actin than control cells (Fig. 3F). The overall phenotype reflects that of macrophages lacking regular Myo9b motor activity[29], which results in major shape changes with a rounded appearance, failures in

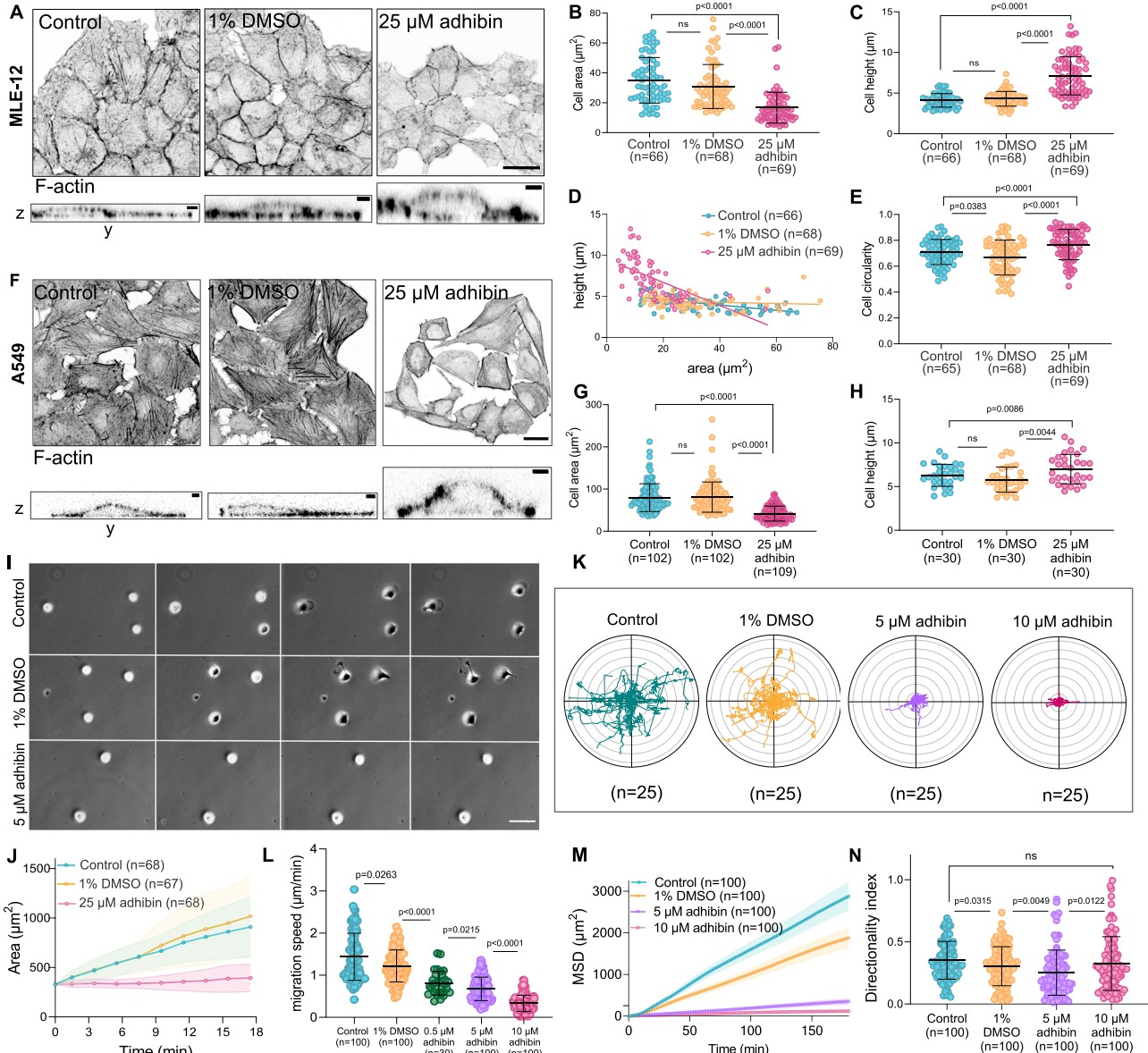

**Fig. 2 | Adhibin causes cell detachment, prevents cell spreading, and inhibits migration. A** Representative *z*-projections (scale bar: 20 μm) and *z*-to-*y* axis views (scale bar: 1 μm) of control, 1% DMSO, and 25 μM adhibin treated MLE-12 cells stained for actin. **B** Cell area (two-tailed unpaired *t*-test, two-tailed Mann–Whitney test for two-column comparison, or one-way Kruskal–Wallis test for multiple comparisons), (**C**) height (two-tailed Mann–Whitney test for two-column comparison, or one-way Kruskal–Wallis test for multiple comparisons), (**D**) height-area correlation, and (**E**) circularity (two-tailed unpaired *t*-test, or one-way ANOVA for multiple comparisons) of control and adhibin treated MLE-12 cells.
**F** Representative *z*-stack projections (scale bar: 20 μm) and *z*-to-*y* stack (scale bar: 1 μm) of control, 1% DMSO, and 25 μM adhibin treated A549 cells stained for actin.

**G** Cell area (two-tailed Mann–Whitney test for two-column comparison, or one-way Kruskal–Wallis test for multiple comparisons) and (**H**) height (two-tailed unpaired *t*-test, or one-way ANOVA for multiple comparisons) of control and adhibin-treated A549 cells. **I** Spreading of B16-F1 cells, control and adhibin treated. Scale bar: 10 μm. **J** Radar plots of single B16-F1 cells in the absence and presence of adhibin. **K** Cell area of B16-F1 cells upon adhibin treatment over time. **L** Migration speed (two-tailed Mann–Whitney test or unpaired *t*-test for two-column comparison), (**M**) mean square displacement, and (**N**) directionality index (two-tailed Mann–Whitney test or unpaired *t*-test for two-column comparison) of control, 1% DMSO and adhibin treated B16-F1 cells. *n* is the number of cells analysed. All data are represented as mean ± S.D.

lamellipodia formation, and a compromised migration ability. Thus, the adhibin-induced defects can be related to deficient Myo9 motors that are unable to properly regulate RhoGAP activity in RhoA-mediated processes driving actin polymerization at the leading edge of cells through which lamellipodia are formed and controlled in their dynamics. To support this conclusion, we investigated the influence of adhibin on actin stability. We performed FRAP experiments and monitored the recovery dynamics of actin polymerization within lamellipodia (Fig. 3G, Supplementary Fig. 4M, and Supplementary Movie 4). Additionally, we analysed total polymeric actin levels by flow cytometry (Supplementary Fig. 3N) and immunoblotting

(Supplementary Fig. 4O, P), including quantification of stress fibres by fluorescence microscopy (Fig. 3H, I). Altered G-to-F-actin ratios and a dose-dependent loss of stress fibres (Fig. 3J, K) support a drug-induced defect in actin polymerization and/or actin filament stability. Contrary to Myo9b depletion, which has been shown to stimulate actin polymerization[29] and stress fibre formation[35], adhibin induces the opposite effects, from which we conclude that adhibin downregulates RhoA activity through the inhibition of Myo9 motor function, which we aimed to validate by pull-down assays quantifying the amount of total vs active RhoA (RhoA·GTP). Adhibin treatment caused a pronounced reduction in active RhoA by approx. 25–95% relative to total

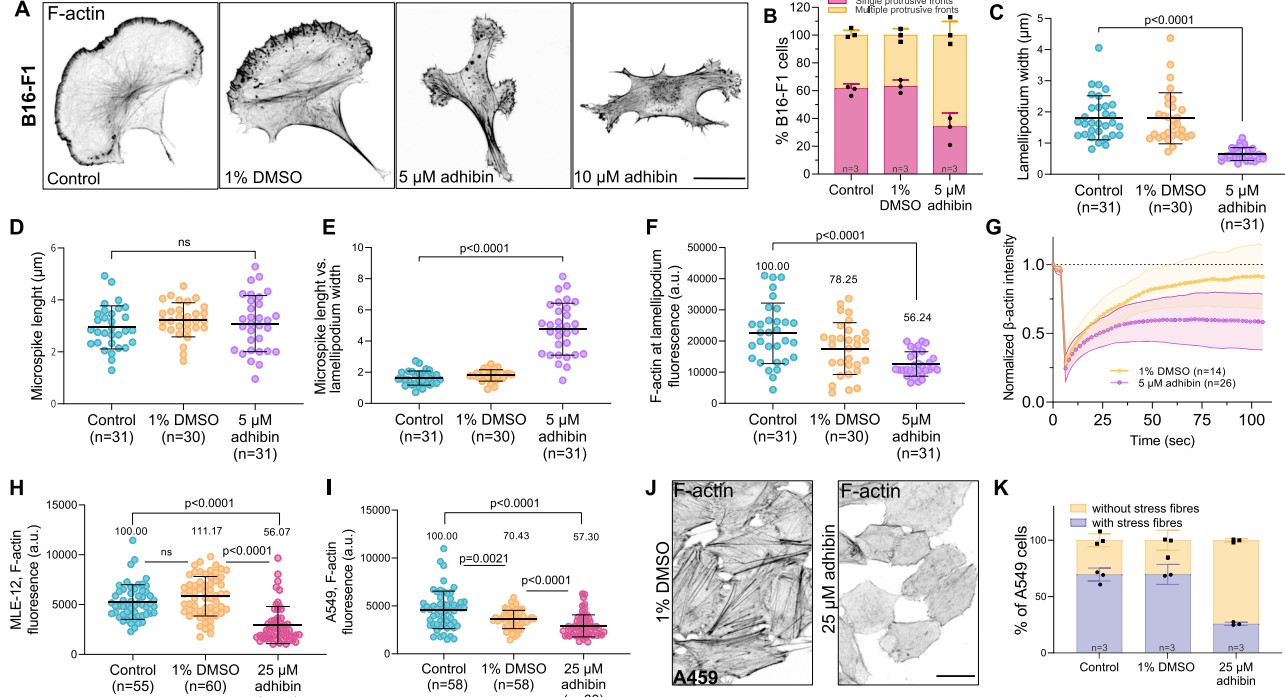

**Fig. 3 | Adhibin perturbs actin dynamics and stability. A** Representative *z*-slice of F-actin in B16-F1 cells before and after treatment with 5 µM and 10 µM adhibin (18 h exposure). Scale bars: 20 µm. **B** Percentage of B16-F1 cells with one or multiple protrusive fronts. **C** Lamellipodium width (one-way Kruskal–Wallis test), (**D**) microspike length (one-way ANOVA), (**E**) microspike length relative to lamellipodium width (one-way Kruskal–Wallis test), (**F**) F-actin content at the lamellipodium (one-way Kruskal–Wallis test) in adhibin treated and B16-F1 cells. **G** Normalized fluorescence recovery curves of β-actin intensity at lamellipodia. **H** Cytoplasmic F-actin content in MLE-12 cells (two-tailed Mann–Whitney test for two-column comparison, or one-way Kruskal–Wallis test for multiple comparisons), and (**I**) in A549 cells with and without adhibin treatment (two-tailed Mann–Whitney test for two column comparison, or one-way Kruskal–Wallis test for multiple comparisons). **J** Representative *z*-projections showing actin in control and 25 µM adhibin treated A549 cells. Scale bars: 20 µm. **K** Percentage of A549 cells with and without stress fibres. In all bar diagrams *n* is the number of experiments and in all box plots n is the number of cells analysed. All data are represented as mean ± S.D. a.u. = arbitrary units.

RhoA levels, which were also reduced (Supplementary Fig. 4Q). This agrees with the results of active RhoA quantified by ELISA (Supplementary Fig. 4R). The reduced pool of active RhoA throughout the various cell lines studied supports a mechanism of RhoA down-regulation. Important hints supporting a mechanism of RhoA down-regulation are the mislocalisation of active RhoA and Myo9b. Active RhoaA visualized with a fluorescent RhoA·GTP sensor (Fig. 4A and Supplementary Movie 5)[68], as well as Myo9b (Fig. 4B–D), both were absent from cortical sites at the cell rear and within protruding cell fronts, and distributed throughout the entire cytoplasm without a distinct preference for a particular cellular compartment or membranous, cytoskeletal association. This apparent misdistribution of both interactors argues for a mechanism, where adhibin affects the proper localization of Myo9b that induces local disturbances in Rho-signalling affecting actin dynamics and stability of actomyosin contractile structures in a similar way as reported for ROCK and RhoGEF inhibitors[69–71]. Since RhoA mediates myosin light chain (MLC) phosphorylation by activating ROCK, the observed loss of phosphorylated MLC (pMLC) from the rear cortex, stress fibres, and actin-rich cortical sites (Fig. 4E) provides additional support for a downregulated RhoA pathway.

Apart from localizing at actin-rich regions of extending lamellipodia and ruffles (Fig. 4F), Myo9b has been shown to accumulate at the tips of filopodia and retraction fibres[72]. Adhibin treatment resulted in a significant loss of endogenous Myo9b motors at filopodia tips and within retraction fibres in NIH-3T3 cells (Fig. 4G, H). This observation indicates that adhibin effectively inhibits Myo9b motor function intracellularly, which prevents self-transportation of the motor to sites of active RhoA for a negative and spatiotemporal control of RhoA-mediated signal propagation. In A549 cells, Myo9b colocalized with paxillin and F-actin patches at focal adhesions (FAs), preferentially at stress fibre tips (Fig. 5A). We confirmed this distinct Myo9 localization additionally in the Calu-3, Caco-2, MLE-12, and HeLa cell lines, implicating a conserved functional role of Myo9b in cell adhesion (Supplementary Fig. 6A). Our data coincide with the reported localization of Myo9b at membrane ruffles and lamellipodia of prostate cancer cells[35], or within FAs of osteoblasts[73] and Caco-2 cells[74], which prompted us to investigate the impact of adhibin on Myo9b function in FAs further by quantifying the cellular distribution of functionally related focal adhesion proteins, including paxillin and vinculin, which both bind actin and act as scaffolding proteins at the early stages of focal adhesion assembly[75], and the vasodilator-stimulated phosphoprotein (VASP), an actin elongator involved in regulating lamellipodium dynamics and cell migration[65,76]. Adhibin treatment led to a significant decrease in the number and area of FAs in a reproducible manner as indicated by the gradual disappearance of the proteins from cortical and surface-associated subcellular structures (Fig. 5A–D and Supplementary Fig. 7A–D); however, without significantly reducing total protein levels (Supplementary Fig. 8A–D). *MYO9B* silencing led to a slight increase in the number and area of focal adhesions and the cells displayed a phenotype characterized by an increased number and size of stress fibres (Supplementary Fig. 7E), which can be attributed to increased RhoA activity due to Myo9b loss-of-function and lack of Myo9b-mediated regulation of RhoA downstream effector signalling, which agrees on previous findings[35]. However, prominent changes in actin and focal adhesion organization were largely missing in Myo9b-depleted cells upon drug

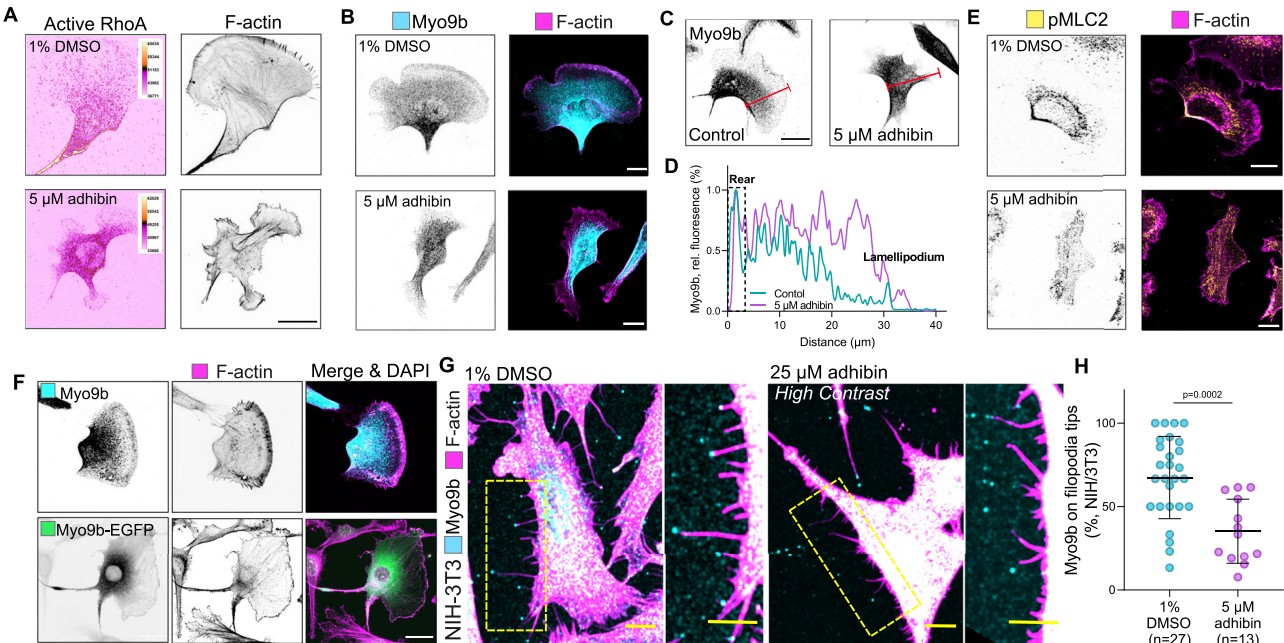

**Fig. 4 | Adhibin causes Myo9b misdistribution and interferes with RhoA-signalling. A** Representative z-projections showing actin and active RhoA (colour-coded dtomato-rothekin intensity, scale bars: 20 μm) and (**B**) Myo9b and actin localization in adhibin-treated B16-F1 cells (scale bars: 10 μm). **C** z-Projections showing localization of Myo9b in B16-F1 cells before and after adhibin treatment. The red line defines the distance from rear to front. Scale bars: 20 μm. **D** Intensity profile of Myo9b along the rear to front section of B16-F1 cells in the presence and absence of adhibin. **E** pMLC2 and actin localization in 1% DMSO and 5 μM adhibin treated B16-F1 cells. Scale bars: 10 μm. **F** z-Projections of native and Myo9b-eGFP-overexpressing B16-F1 cells showing Myo9b and actin localization. Overexpression of myo9b affects lamellipodium architecture and cell shape. Scale bars: 20 μm. **G** Representative z-projections of NIH-3T3 cells showing Myo9b localization at tips of filopodia and retraction fibres. Scale bars: 10 μm. **H** Percentage of Myo9b at tips of filopodia and retraction fibres in NIH-3T3 cells with and without adhibin (two-tailed unpaired t-test). n is the number of cells analysed. All data are represented as mean ± S.D. All experiments were reproduced at least three times with similar results.

treatment (Supplementary Fig. 7F–H). These absent effects in Myo9-silenced cells provide additional support for Myo9 being the functional target of adhibin.

Experiments with the ROCK inhibitor Y-27632, which acts downstream of RhoA, induced comparable defects, implying that adhibin and Y-27632, both target the same signalling pathway. Combined adhibin and Y-2763 treatments led to a similar loss of FAs as single drug administration (Fig. 5C, D). Correlation analysis of the paxillin and Myo9b signals revealed reduced Myo9b levels at FAs (Fig. 5E). Upon adhibin wash-out, cells reacquired their typical morphology, formed lamellipodia, stress fibres, and dynamic FAs. These findings reveal that adhibin affects the dynamics of Myo9b at adhesive structures and weakens its association at the tips of stress fibres. Together, the data indicate that adhibin prevents Myo9b self-transportation at adhesion sites of active actin polymerization[27,72].

Since adhesion is closely associated with the molecular composition and structure of the extracellular matrix (ECM)[77], we analysed the effect of adhibin on cell-matrix adhesion, exemplary in A549 cells using different surface coatings, including fibronectin, laminin, and PLL. Cells constantly failed to establish stable surface attachments, which were independent of the coating and progressed in a concentration-dependent manner (Supplementary Fig. 6B–F). To corroborate the assumption that adhibin might interfere with the assembly, maintenance, and/or structural integrity of FAs, we performed live-cell TIRF-microscopy with A549 cells expressing eGFP-paxillin and analysed FA dynamics. Compared to control cells, adhibin-treated cells displayed reduced FA movements (Fig. 5F, G) and FA decomposition advanced considerably faster than in control cells as revealed by autocorrelation analysis (Fig. 5H). Collectively, the data indicate that adhibin causes a reduction in the number, area, and dynamics of FAs, which appears to be insufficient for the cells to maintain a stable and firm connection with the underlying matrix.

Including that RhoA-activated NM2 function is indispensable for the maturation of FAs[78,79], we addressed the role of spatiotemporal active RhoA and NM2A/B on the stability of the focal complex[70]. For this purpose, we studied the localization of phosphorylated myosin light chain (pMLC2) in co-stainings with phalloidin to visualize active actomyosin contractile structures that determine FA adhesion dynamics[79] and enable cell migration[66]. Active pMLC2 was absent from actin fibres and did not localize within cortical regions (Fig. 6A). Cells increasingly lacked transverse arcs and dorsal stress fibres (Fig. 6B) and the levels of pMLC2 were significantly reduced (Fig. 6C–H), indicating suppression or partial inactivation of the contractile actomyosin machinery. The data reveal that adhibin targets primarily the RhoA-signalling pathway that is responsible for regulating actin polymerization and actomyosin contractility, both determining the adhesive, contractile, and motile properties of cells. This conclusion is further strengthened by the loss of the cortical association of both NM2 isoforms in adhibin-treated cells (Fig. 7A, B). Since total NM2A and NM2B levels were not affected (Fig. 7C–F), the absence of NM2A/B from cortical and contractile cell structures can be interpreted by a suppressed pathway of MLC2 phosphorylation required for activating NM2. Concomitantly, we detected a significant reduction in the levels of phosphorylated myosin light chain phosphatase (MLCP) by adhibin (Fig. 7G), which provides additional support of a down-regulated RhoA/ROCK pathway, where both, inactive MLC and active MLCP synergistically inactivate NM2. Such a mechanism could explain the observed defects in single and collective cell migration, where deactivated NM2 motors are unable to stimulate cell contractility or contribute to the stable formation of stress fibres. We interpret this as a consequence of Myo9b motors being inhibited in their ability to perform directional movements, which prolongs the residence time at sites of active RhoA, locally leading to a reduction in the pool of active RhoA required for stress fibre formation and activation of contractility.

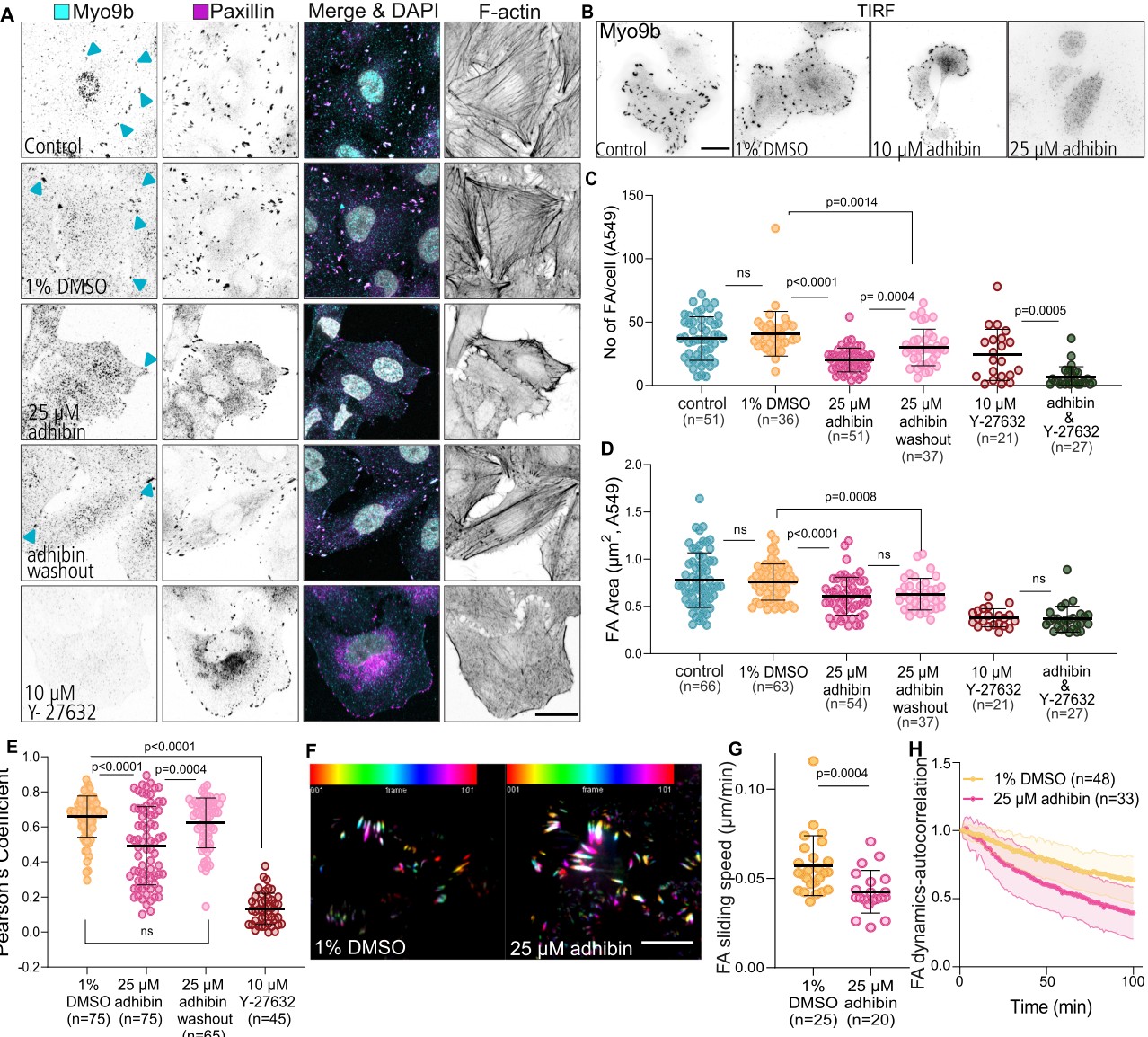

**Fig. 5 | Adhibin impairs lamellipodia and focal adhesion dynamics.**
**A** Representative *z*-slice showing Myo9b, paxillin, and actin localization in A549 cells cultured for 18 h in a medium supplemented with 1% DMSO, 25 µM adhibin, or 10 µM Y-27632. Scale bars: 20 µm. **B** Representative TIRF acquisitions showing Myo9b in A549 cells with and without adhibin. Scale bars: 20 µm. The experiment was reproduced three times with similar results. **C** Quantification (two-tailed Mann−Whitney test) and (**D**) size of focal adhesions (FA) (two-tailed Mann−Whitney test or unpaired *t*-test) before and after treatment with adhibin and Y-27632.

**E** Pearson's coefficient of Myo9b and paxillin colocalization (two-tailed Mann−Whitney test). *n* = number of FA. **F** 2D time series showing EGFP-paxillin signals in A549 cells over 90 min with and without adhibin treatment. Colour coding defines the frame number. White colour shows stable signals over time. Scale bars: 5 µm. **G** FA sliding speed in the absence and presence of 25 µM adhibin (two-tailed Mann−Whitney test). **H** Representative auto-correlation curves of FA dynamics in eGFP-paxillin transfected A549 cells with 1% DMSO and 25 µM adhibin. *n* is the number of cells analysed. All data are represented as mean ± S.D.

Analysis of cytoskeleton-associated genes including *RHOA*, *MYO9A*, *MYO9B*, *PXN*, *VCL*, *VASP*, *ACTβ*, *MYH9*, as well as a selection of proto-oncogenic and tumour-suppressive targets related to drug anti-metastasis, such as *BRAF*, *CDK4*, *EGFR*, *KEAP1*, *KRAS*, *PIK3CA*, *PTEN*, *PTPRD*, *TP53*[80–84] revealed no significant changes in expression levels, except for *PXN*, *VASP*, and *KEAP1* (Fig. 7H−K). Upregulated *PXN* and *VASP* suggest a feedback response of the cells to compensate for the deregulated Rho-signalling and to account for the defects in the structural/mechanical links between the ECM and the actin cytoskeleton[85] induced by adhibin. On the other hand, downregulated *KEAP1* expression correlates with the phenotype of suppressed cell migration and invasion, causing cell shrinkage due to decreased focal adhesions via an inhibited RhoA-ROCK pathway, as demonstrated by the KEAP1/NRF2 inhibition in non-small-cell lung cancer cells[86] as a strategy for treating cancers with high NRF2 activity[87].

The interfering effect of adhibin with RhoGTPase signalling becomes also apparent from failures in cell division. RhoA is essential for defining the sites that initiate ingression of the contractile furrow during cytokinesis[88]. Restricted spatial regulation of RhoA and RhoA-controlled citron kinase activity can lead to failures initiating membrane ingression and midbody formation[88,89]. Adhibin-treated B16-F1 cells increasingly failed to accomplish cytokinesis (Fig. 7L−O and Supplementary Movie 6). Since RhoA coordinates cytokinesis by directly activating mDia2-mediated F-actin assembly, stimulating further NM2 via ROCK-mediated MLC2 phosphorylation, which promotes the assembly and constriction of the contractile ring[88], the observed mitotic defects support the conclusion that adhibin targets this pathway through a Myo9-mediated down-regulated RhoA pathway. This conclusion is further strengthened by the partial absence of pMLC from the actin contractile ring (Fig. 7M, N). With adhibin, the

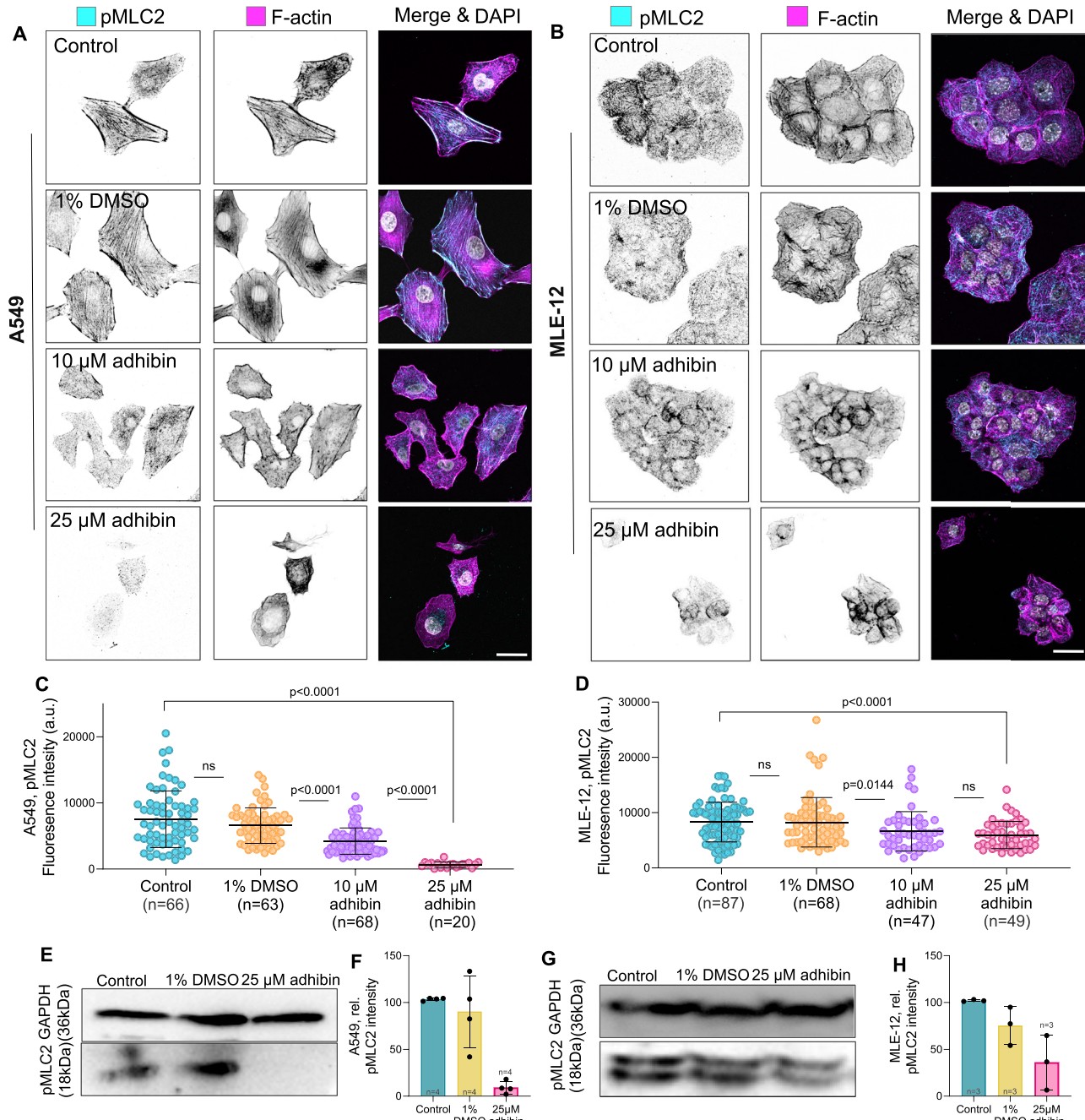

**Fig. 6 | Adhibin interferes with RhoA-mediated activation of NM2 contractility.** **A**, **B** Representative sum-projections showing actin and pMLC2 localization in adhibin-treated A459 and MLE-12 cells. Scale bars: 20 µm. **C**, **D** Amount of pMLC2 in control and adhibin-treated A459 cells, and MLE-12 cells (two-tailed Mann–Whitney test for two-column comparison, or one-way Kruskal–Wallis test for multiple comparisons). **E**, **G** Western blots comparing the amount of pMLC2 between adhibin-treated and A549 and MLE-12 cells relative to GAPDH. **F**, **H** Normalized pMLC2 intensity from three individual western blots. In all bar diagrams *n* is the number of experiments and in all box plots *n* is the number of cells analysed. All data are represented as mean ± S.D. a.u. = arbitrary units.

percentage of cells containing multiple nuclei was significantly higher than in control cells (Fig. 7O). Since adhibin does not target any of the three NM2 isoforms (Fig. 1F), the observed defects in migration and cytokinesis can be explained by a mechanism of RhoA inactivation and not through an off-target effect on NM2 function. The data allow the conclusion that loss of NM2-activity is related to a RhoA-downregulated pathway of MLC activation.

### Adhibin disassembles cell−cell contacts by disrupting cell−cell junctions

Apart from the prominent role of Myo9b in cell adhesion and migration, the Myo9a isoenzyme has been shown to actively participate in

the assembly of actin bundles[90] at nascent cell−cell adhesions[31]. Adherence junctions and tight junctions provide the adhesive contacts between neighbouring cells[91]. Adhibin caused loss of cell−cell connections (Fig. 2A and Supplementary Fig. 1M), presumably by promoting the disintegration of tight-junctions through RhoA-related mechanisms of actin depolymerisation and loss of Myo9a motor activity preventing the motors from maintaining the scaffolding and actin-crosslinking activity in a spatiotemporally controlled manner.

To study the effect of adhibin on cell−cell contacts and test for a direct interference with the capability of Myo9a to actively contribute to the stability of cell cohesive structures and the RhoA signalling cascade regulating cell−cell junction architecture and integrity[31], we

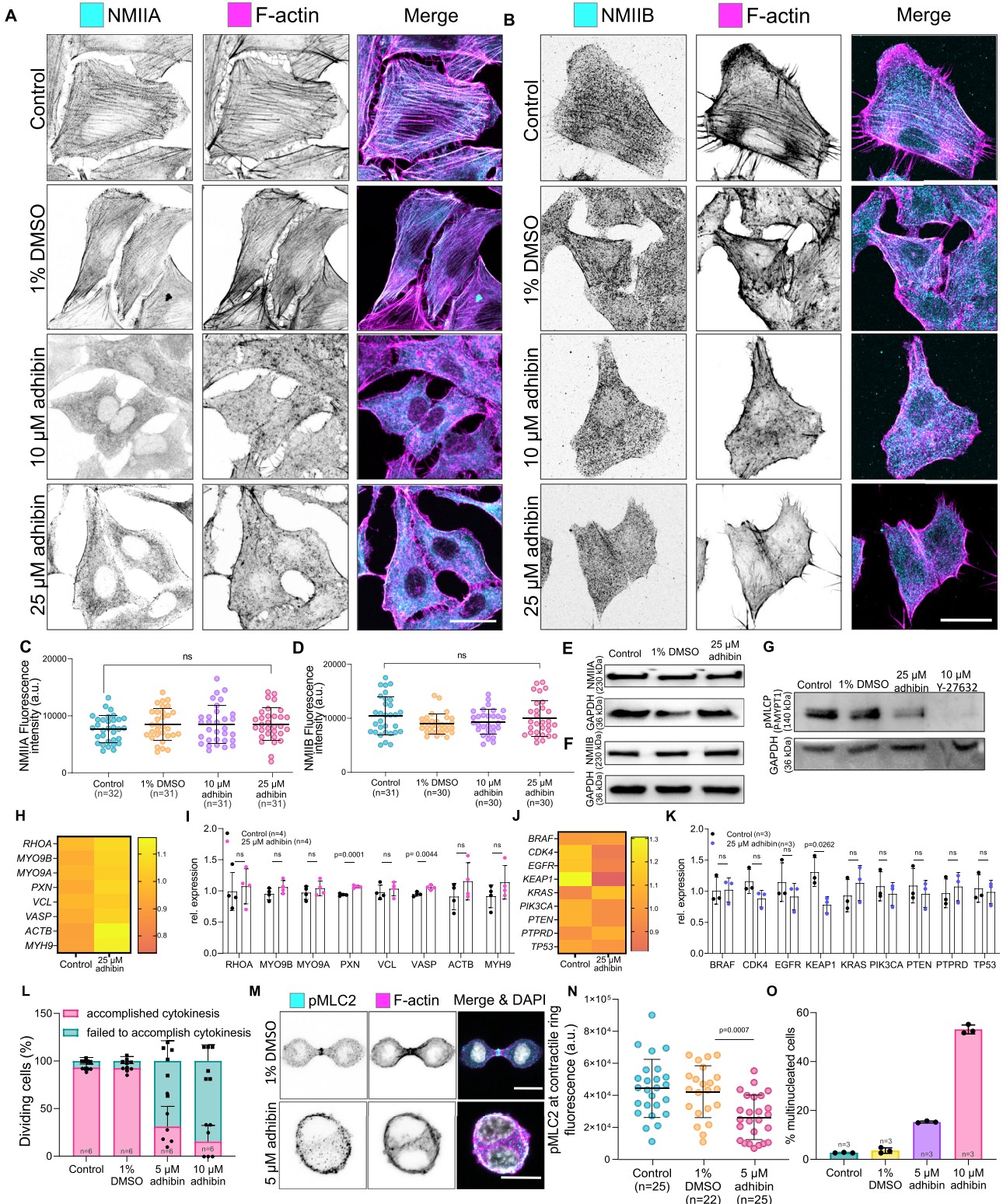

used the MLE-12 cell line, which is characterized by the formation of a tight epithelial layer. Myo9a displayed a clear colocalization with actin along the membranes of neighbouring cells (Fig. 8A). Upon adhibin treatment, cells subsequently lost connections with the surrounding cell neighbours. The further the separation process progressed, the less Myo9a was associated with cortical actin along the cellular contacts (Fig. 8A). Longer drug exposures caused the complete disassembly of the epithelial monolayer, where single cells detached completely from the surface. The tight junction protein ZO-1, which

connects the actin cytoskeleton to cell–cell membrane junctions[92], also lost its membranous association. It distributed in speckles throughout the entire length of the membranes (Fig. 8B) and displayed the same punctured architecture as cortical actin (Fig. 8C). Analysis of the z-axis intensity of the cells highlights the disturbed localization of ZO-1 after adhibin treatment (Fig. 8D), whereas total levels remained almost unaffected (Supplementary Fig. 8E, F). Accordingly, we observed the dislocation of the gap-junction protein connexin-43[93] (Cx43) from the membrane and its complete disappearance at 10 μM

**Fig. 7 | Adhibin prevents stress fibre formation and interferes with cytokinesis by downregulating RhoA pathways of NM2A/B activation. A, B** Representative confocal images showing NM2A and NM2B localization and loss of stress fibres in phalloidin stained A459 cells upon adhibin treatment. Scale bars: 20 μm.
**C, D** Amount of NM2A and NM2B in control, 1% DMSO, 10 μM adhibin and 25 μM adhibin treated A459 cells (C: one-way ANOVA, D: one-way Kruskal–Wallis test).
**E, F** Western blots of A549 whole cell lysates showing the total amount of NM2A and NM2B in adhibin treated and cells relative to GAPDH. **G** Western blot of A549 whole cell lysates showing the total amount of p-MYPT1 in adhibin and Y-27632 treated and cells relative to GAPDH. **H, K** qPCR data showing relative changes in the expression of cytoskeleton-associated genes (**H**–**I**: two-tailed unpaired *t*-test) and

metastasis-related genes (**J, K**: two-tailed unpaired *t*-test or Mann–Whitney test) upon adhibin treatment as heat map and bar diagram representation. **L** Percentage of B16-F1 cells entering mitosis during 3 h. **M** Confocal *z*-projections of pMLC2 and phalloidin in dividing B16-F1 cells in the presence and absence of adhibin. Scale bars: 10 μm. **N** Fluorescence intensity of pMLC2 at the contractile ring of dividing B16-F1 cells in the presence and absence of adhibin (two-tailed unpaired *t*-test). **O** Percentage of B16-F1 cells with multiple nuclei with and without adhibin addition. In all bar diagrams *n* is the number of experiments and in all box plots *n* is the number of cells analysed. All data are represented as mean ± S.D. All experiments were reproduced at least three times with similar results. a.u. = arbitrary units.

and 25 μM drug concentrations (Fig. 8G). This suggests that the structural integrity and function of gap junctions could be affected by adhibin. Transepithelial electrical resistance (TEER) measurements using impedance spectroscopy in colon-derived epithelial Caco-2 cells, which form even more tight epithelial layers than MLE-12 cells, highlight adhibin-induced defects in cell junction integrity (Fig. 8E). One-time drug application rendered the barrier leaky over days (Fig. 8F). Cells exhibited disrupted tight junction patterns visible in the fragmented arrangement of ZO-1 along neighbouring membranes and in the irregular cortical F-actin organization (Fig. 8E), comparable with the patterns previously observed with MLE-12 cells.

Our findings indicate that adhibin disturbs cell–cell connections, by affecting both, tight junction stability and gap junction integrity. Previous studies already showed that knocking down Myo9a and Myo9b in epithelial cells resulted in the loss of ZO-1 and tight junction disruption[38,94], emphasizing that adhibin targets the RhoA signalling cascades regulating the formation, maturation, and maintenance of cell junctions.

## Adhibin abolishes tumour cell migration and retards organoid morphogenesis without causing animal lethality or pathological effects

In order to test the drug for anti-metastatic properties, we used spheroids. Compared to assays performed with tumour cells cultured in medium on a flat surface, spheroid models based on immortalized cell lines are more closely related to the complex nature of a tumour, mimicking its three-dimensional (3D) assembly and growth[95]. Spheroids provide an ideal system for drug testing approaches prior to animal experiments. We therefore chose the NIH-3T3 embryonic fibroblast cell line as a spheroid model to study the effect of adhibin on metastasis-related features of cells, including 3D migration and multicellular assembly. Within a confined MATRIGEL environment, adhibin completely abolished the ability of the cells to separate from the 3D assembly and migrate under all tested concentrations (Fig. 9A–C). We also studied the formation of spheroids from single cells (Fig. 9D, E) revealing significant retardation of the assembly process at 5 μM and 10 μM adhibin concentrations, which was completely suppressed at higher doses (Fig. 9F).

To study the consequences of adhibin administration on complex cellular structures as present in tissue and organs, we isolated and cultured intestinal organoids, three-dimensional multi-component cell models, which are capable of representing the morphogenesis of intestinal crypt-villus architecture in vitro[96]. Over-expansion of intestinal organoids in culture, causes proliferative zones harbouring the stem cells to gradually develop into crypt-like structures, which bud out and acquire eccentricity in the originally spherical organoid[97,98]. This morphogenesis is essentially controlled by NM2. In intestinal crypts and crypt-like structures of the intestinal organoids, NM2 piles up at the apical side of the cells, at the base of the crypts, and at the basal side of the cells at the crypt-villus wall. This differential localization of NM2 provides a crypt apical contraction and villus basal tension to support the formation and elongation of the crypts, which is disturbed by blebbistatin[97–101]. Adhibin treatment of intestinal

organoid cultures from human mid-colon biopsies[102] started on day 2 of culture reduced the prevalence of the budding events in the colonoids on the following days and resulted in decreased eccentricity in the colonoids in comparison to the control and vehicle group (Fig. 9G, H). Growth (as determined by measuring cross-section area over time) or survival of the colonoid were insignificantly influenced by adhibin treatment (Fig. 9I, J). Comparable mRNA expression of the intestinal proliferative cell marker, *Ki-67*, among all samples suggests that adhibin-induced changes in colonoid morphogenesis occur independently from their regenerative capacity (Fig. 9K), i.e. the proliferative cells are available, but not able to bud into crypt-like protrusions. The mRNA expression of *MYO9A* and *MYO9B* was also comparable among different groups (Fig. 9K). Collectively, these data show that adhibin can be tolerated by colonoids.

To assess potential off-target effects independent of Myo9 as a target and to test for organismal toxicity, we took *D. melanogaster* as a representative animal model, which lacks class-IX myosins[103]. Adhibin was administered at doses up to 100 μM and uptake was confirmed in adult flies and larvae by mass spectrometry (Supplementary Table 3, 4). The drug caused no visible effects on the morphology of the adult flies. Both female and male adult flies and larvae exhibited normal fertility and growth (Supplementary Fig. 9A–C). To test whether adhibin has an effect on the RhoGTPase signalling independent of Myo9, we examined the phenotypes previously described to be associated with the disruption of this signalling pathway[104–106]. Progeny from adhibin-treated parents showed normal nervous system development during embryogenesis (Supplementary Fig. 9D) and photoreceptor axonal projections were properly established in the brain of adhibin-treated larvae (Supplementary Fig. 9E). Additionally, in adult brains, there were no apparent changes observed in the structure of the learning and memory centre or the tight junctions of the blood-brain barrier[107] (Supplementary Fig. 9F–G).

In mice, intraperitoneal administration of adhibin, which displays high solubility up to 25 μM concentrations (Supplementary Fig. 10A), produced minor compound-related effects on body weight (Supplementary Fig. 10B) and no animal lethality within a period of 14 days after administration (Supplementary Fig. 10C). The general health status of the mice, as indicated by the clinical scoring system, was unaffected (Supplementary Fig. 10D). The histopathological evaluation of organs and bone marrow, after evaluating drug distribution in blood serum (Supplementary Fig. 10E), revealed no significant effects (Supplementary Fig. 10F). Focal acute fibrinoid changes with minor acute haemorrhage in the lungs of adhibin-treated mice without cellular inflammation was found euthanasia-associated[108]. A single intraperitoneal injection of adhibin up to 0.142 mg/kg body weight was tolerated by the mice without chronic organ damage, providing the initial concentration range for the therapeutic evaluation of adhibin in in vivo models of solid and disseminated tumours.

## Discussion

The concept of targeting RhoGTPase signalling pathways has evolved as a promising strategy for treating metastatic cancers. Progress has been made with a series of chemical compounds that are capable of

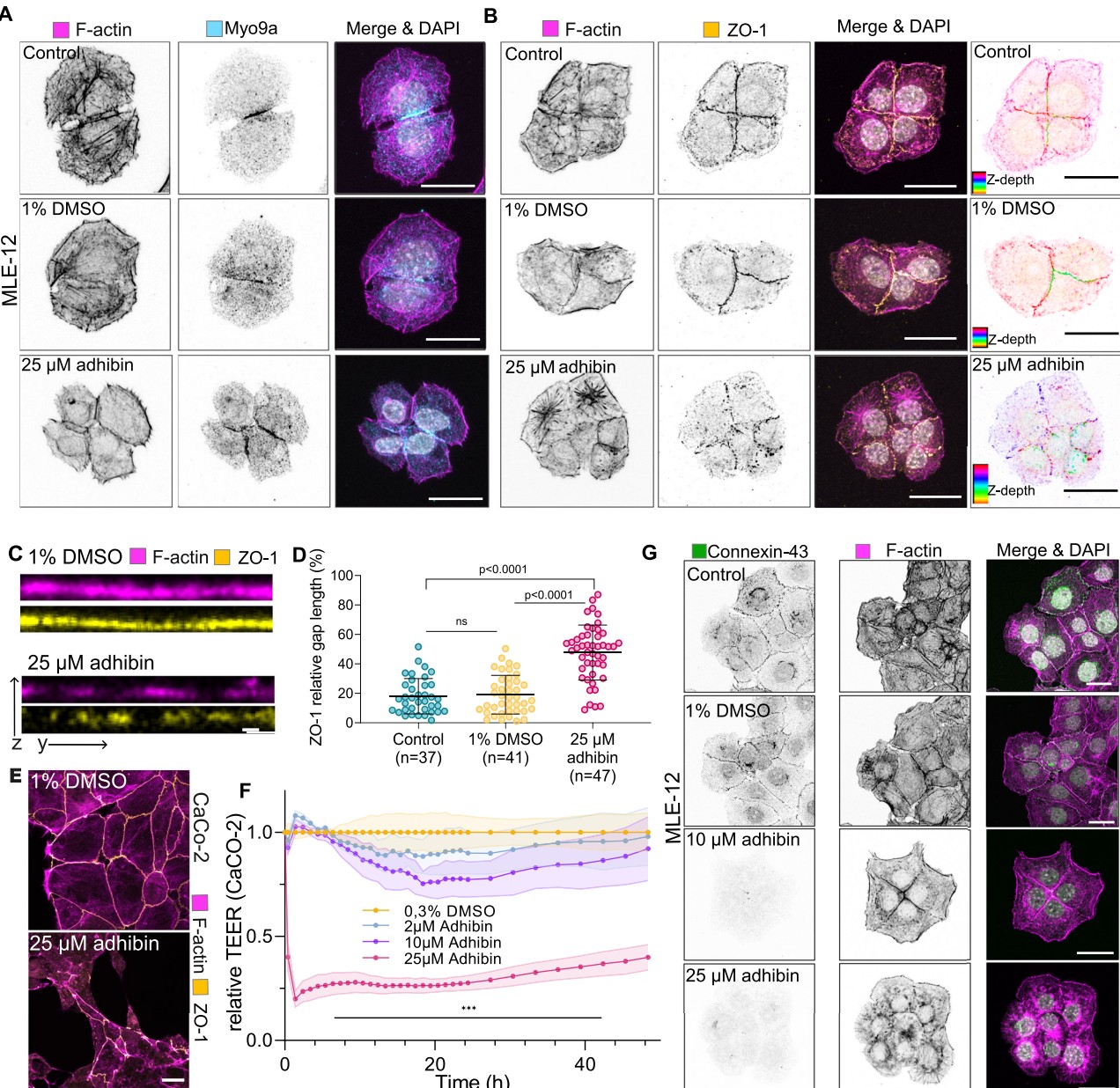

**Fig. 8 | Adhibin disassembles cell–cell contacts and disrupts tight junction integrity. A** Representative z-projections showing Myo9a and actin localization, (**B**) ZO-1 and actin localization in control, 1% DMSO and 25 μM adhibin treated MLE-12 cells. ZO-1 spatial distribution is also shown as colour-coded z-depth projections. Scale bar: 20 μm. **C** ZO-1 and actin intensity profile of cell–cell contact area in control and adhibin-treated MLE-12 cells. Scale bars: 2 μm. **D** ZO-1 relative gap length along the cell–cell contact area in control, 1% DMSO and 25 μM adhibin treated MLE-12 cells (unpaired t-test for two-column comparison, or one-way ANOVA for multiple comparisons), n is the number of cells analysed. **E** Representative confocal z-projections showing ZO-1 and actin localization in 1% DMSO and 25 μM adhibin-treated CaCo-2 cells. Scale bar: 20 μm. **F** Relative TEER (transepithelial electrical resistance) of Caco-2 cells. Shaded areas indicate the amplitude of variability around the mean of 3 experiments. **G** Representative confocal z-projection showing localization of connexin-43 and actin in control, 1% DMSO and 25 μM adhibin treated MLE-12 cells. Scale bars: 20 μm. Data are represented as mean ± S.D. All experiments were reproduced at least three times with similar results.

interfering with all modes of cancer cell invasion and metastasis, generally referred to as migrastatics[19]. These structurally distinct compound classes target downstream effector mechanisms underlying the pro-migratory, invasive, and/or highly proliferative features of cancer cells, either by disturbing stability and dynamics of the actin cytoskeleton or by inhibiting actomyosin-based contractility through direct binding to actin, to non-muscle class-II myosins, or to their regulators, or by interfering with (i) the spatial organization of single RhoGTPases, (ii) the binding and exchange of nucleotides to RhoGT-Pases, or (iii) kinase effector-mediated downstream signalling[46]. Despite the anti-migratory and anti-proliferative effects of the

compounds, mainly validated in cancer cell models, only a few have progressed into clinical trials[19]. This stresses the need for pharmacologically more potent substances that elicit anti-metastatic effects through novel targets and/or novel mechanisms. Here, following a chemistry-assisted, mechanism-based drug discovery approach, in combination with X-ray structure analysis, molecular modelling, cancer cell models and cell biological investigations, we have identified a potential anti-metastatic lead termed adhibin that develops its efficacy by targeting the motor function of the RhoGAP class-IX myosins. Adhibin binds to a cryptic pocket that is different from those identified for other myosin effectors[52,53,57,58,109]. It is located at the interface

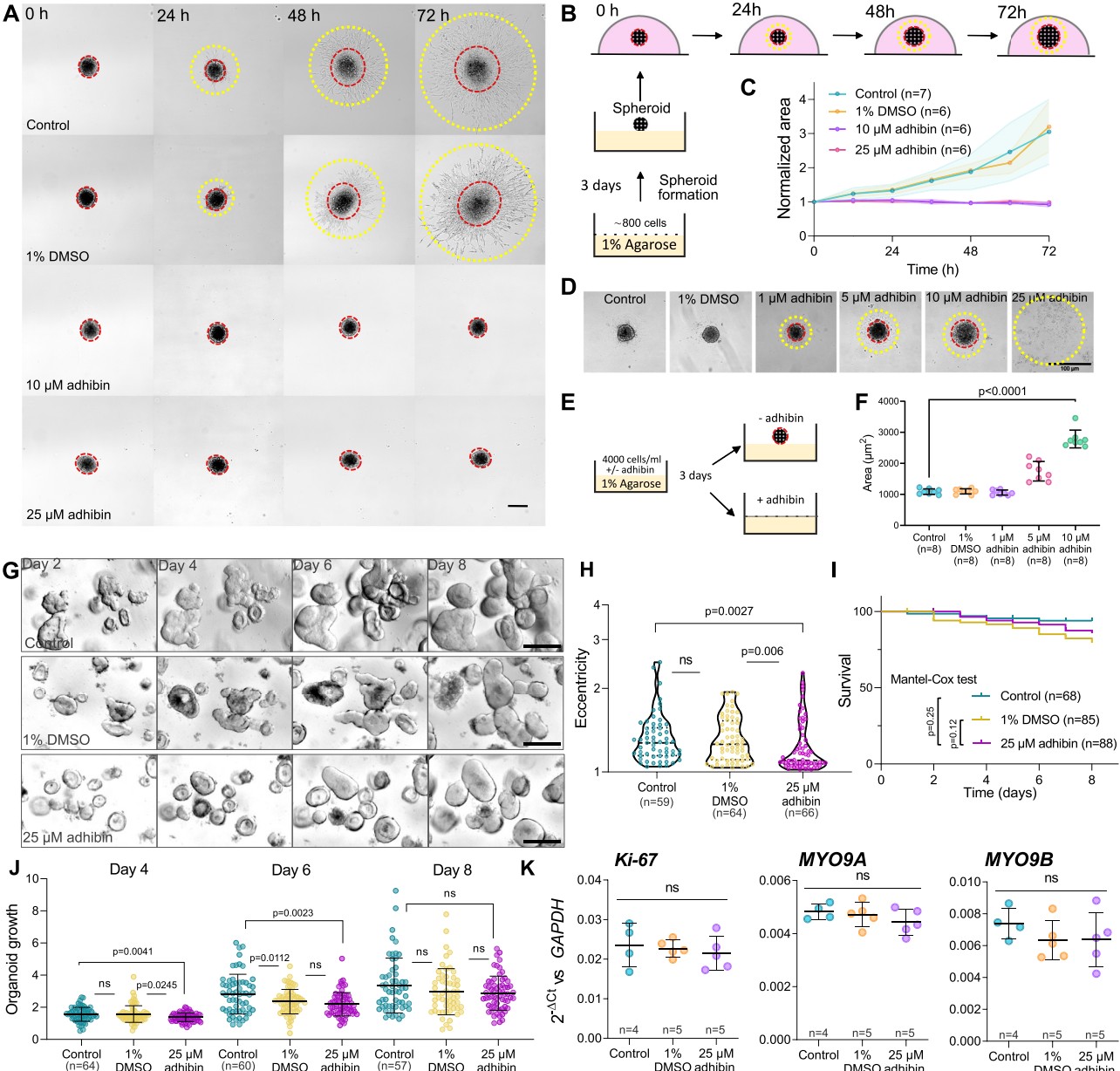

**Fig. 9 | Adhibin prevents 3D migration and spheroid formation and retards the growth and proliferation of human colonoids. A** 2D (yellow circle) and 3D migration (red circle) of 3D NIH-3T3 cells clumped in spheroids during 72 h in the presence and absence of adhibin. Scale bar: 200 μm. **B** Scheme describing the process of the spheroid formation, treatment and imaging. **C** Quantification of NIH-3T3 spheroid area during 72 h in the presence and absence of adhibin. *n* is the number of experiments. **D** Formation of NIH-3T3 spheroids in the presence and absence of adhibin. Scale bar: 100 μm. The experiment was reproduced eight times (see F for quantification). **E** Scheme describes the process of spheroid formation in the absence and presence of different adhibin concentrations. **F** Quantification of NIH-3T3 spheroid area formed (**D**) in the presence and absence of adhibin (one-way

ANOVA), *n* is the number of experiments. **G** Representative images of colonoids at days 2-8 with different treatments. Scale bar: 10 μm. **H** Characterization of colonoid shapes for different conditions at day 6 based on eccentricity (1/circularity) (one-way ANOVA for multiple comparisons, two-tailed Mann−Whitney test for two-column comparison), *n* is the number of cells analysed. **I** Survival rate of colonoids. **J** Proportional growth (as determined by measuring cross-section area over time) of colonoids for each condition at days 4, 6, and 8 (two-tailed unpaired *t*-test or Mann-Whitney test), *n* is the number of cells analysed. **K** RT-qPCR analysis of colonoid cultures at day 6 after different adhibin treatments (one-way ANOVA), *n* is the number of experiments. All data are represented as mean ± S.D.

between the phenamacril and blebbistatin binding sites and transiently formed upon binding of adhibin. Blebbistatin and phenamacril prevent efficient force production by blocking closure of the actin-binding cleft, thereby arresting the motor in a pre-power stroke state of low actin affinity[110,111], whereas adhibin appears to exert its inhibitory potency by impairing conformational transitions of myosin associated with weak-to-strong actin interaction[50], in favour of the strong actin bound states, which would explain both, the reduced motility and the increased fraction of immotile actomyosin states.

Apparently, this allosteric mechanism of inhibition translates in the cellular context into changes of total RhoA levels, as well as local active RhoA fractions, both affecting dynamics of cell membrane protrusions, formation and maintenance of cell-adhesive structures, cell polarity and cell body contraction, contractile ring formation, as well as cell−cell junction integrity. Mechanistically, we interpret the drug-induced cellular defects as a consequence of an intrinsically disturbed Myo9 function: the ability to spatiotemporally control Rho-signalling through a motorized signalling mechanism and scaffolding

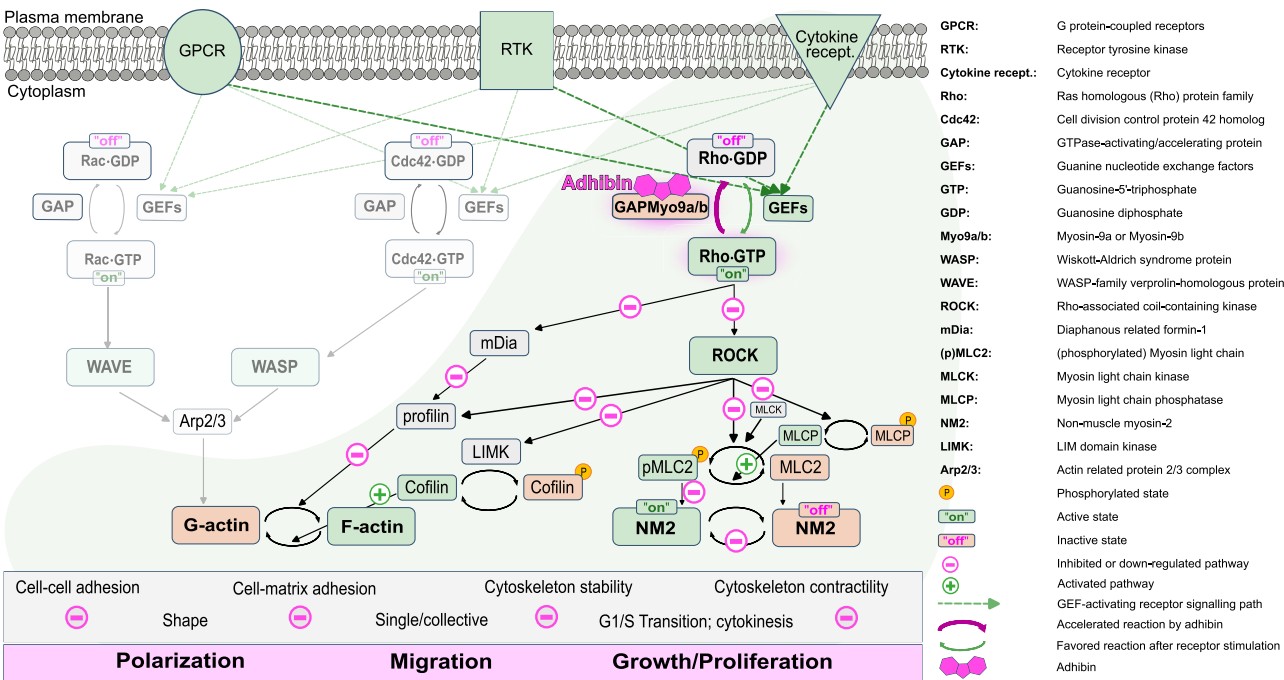

**Fig. 10 | Effect of adhibin on RhoGTPase signalling.** Schematic representation of Myo9-regulated Rho-signalling and pathways affected by adhibin.

activity, which allows for tethering the signalling cues and cytoskeletal components into complexes that regulate the mechanical properties of cells to enable migration, establish cell–cell and cell surface connections, and perform cytokinesis. As highlighted in Fig. 10, the RhoGTPase subfamily members, particularly RhoA, are collectively involved in cell polarization, migration, and proliferation. By activating ROCK[112,113], they control LIMK1/2 and cofilin activity in a reciprocal manner and thus contribute to the assembly and stability of filamentous actin structures[114], whereas Rac- and Cdc42-associated pathways participate in actin filament stability, regulate branching and actin remodelling[115]. Additionally, ROCK activation by RhoA stimulates NM2 contractility and promotes cell retraction through MLC phosphorylation and inactivation of MLCP[116,117]. On the other hand, Myo9b recruitment to membrane extensions by Rac-induced actin polymerisation is assumed to locally inhibit RhoA activity at the cell leading edge, preventing contractility, but stabilising a positive feedback loop that supports protrusion formation required for coordinated migration[28]. Our data argue for a locally downregulated RhoA activity by adhibin-compromised Myo9b motors through a mechanism, where adhibin prolongs the residence time of the motors at sites of active RhoA, causing local disruption of RhoA signalling. This suppressed signal propagation disturbs actin dynamics and actomyosin-based cell contractility. Currently, it is not known whether the Myo9-RhoGAP domain is additionally constitutively activated by adhibin or regulated by additional mechanisms. However, RhoGAP and motor activity could be regulated in a coordinated manner. Inhibition of Myo9 motor activity by adhibin could induce a conformation that favours constitutive GAP activity of the myosin. As a consequence, the prolonged residence of Myo9 with RhoA may locally decrease the pool of active RhoA•GTP. Thus, local RhoA downregulation and globally enhanced Myo9 RhoGAP activity by adhibin, both would explain the observed cellular defects.

This proposed mechanism of action is expected to have several effects: (i) a reduction in the active pool of NM2, due to reduced levels of pMLC, compromises the contractile properties of the cells causing defects in migration[28,73] and cytokinesis/cell cycle failures; (ii) the activation of cofilin and concomitantly the inactivation of formin/profilin trigger actin depolymerisation and perturb the stability of

filamentous actin structures[118–120]. Additionally, since non-muscle class-II myosins have been implicated in promoting cancer cell invasion by maintaining cell contractility[121], adhibin owns key features to abrogate the migratory properties of tumour cells and thus suppress or even prevent tumour dissemination by affecting NM2 contractile activity through a deregulated RhoA pathway. Moreover, since adhibin acts on cytokinesis and cell cycle progression by interfering with NM2-mediated and actin-based contractile ring closure through a deregulated Rho-mediated cell-cycle control[122], it could hinder cancer cells from proliferating fast, thus delaying or even suppressing tumour dissemination.

Besides NM2, which generates the tensional forces of focal adhesions[123], class-IX myosins appear to be directly involved in controlling adhesion dynamics, both between cells and the substratum[35,124]. Since adhibin does not target any of the three NM2 isoforms directly, but impairs primarily the targeting mechanism of class-IX myosins to adhesion sites and membrane protrusions, or in the case of Myo9a to cortical membrane junctions, the defects can be related to impaired self-transportation and additionally to compromised tethering functions. This is supported by the similarities of the phenotypes observed with e.g. hydrolysis defective Myo9b mutants, which are unable to properly position the RhoGAP activity at sites of active Rho for regulation[72]. That adhibin affects cell-matrix adhesion through Myo9b inhibition is also supported by findings, showing that the overexpression of the Myo9 tail without the motor domain causes substantial disruption of cell adhesion structures due to constitutive down-regulation of Rho[73], as we observed with adhibin. Moreover, locally downregulated RhoA signalling by adhibin appears to be sufficient to affect the balance between signal propagation via the Rho and Rac cascades as a prerequisite for fine-tuning actin polymerization at extensions and actomyosin contractility at the sides and rear of migrating cells, mechanisms that are critical for cell extravasation and migration[125].

Cell–cell junctions are controlled in their formation and stability by the Rho/ROCK pathway. Particularly, when lamellipodia of neighbouring cells collide, Myo9a has been implicated in targeting the membrane at nascent cell–cell contacts, through its actin-binding and motor properties, where it is assumed to down-regulate Rho/ROCK

activity at the cell junctions to maintain lamellipodia overlap and control radial actin finger formation during early stages of junction formation[34]. On the other hand, Myo9a RhoGAP activity needs to be temporally removed to enhance Rho activity and enable the maturation of junctional contacts. Thus, adhibin appears to target this class-IX isoform as well, causing disruption of cell–cell connections. This mechanism of interference could resolve solid tumours and prevent growth and colonization as exemplified in vitro with the spheroid model. In summary, the proposed mechanism of action through which adhibin impairs class-IX myosin function is in agreement with the functional defects in cell migration, lamellipodia formation, FA dynamics, stress-fibre formation, cell barrier function, and cytokinesis.

Tumour models have shown that blockade of Rho signalling can suppress the metastatic behaviour of cancer cells, but not the accompanying resistances observed for drugs acting on downstream effectors of the cascade[46]. Here, we have achieved an early, upstream interference of the RhoA signalling cascade by selectively impairing the motor function of class-IX myosins, concomitantly enhancing GAP activity, which suggests local termination of constitutively active RhoA signalling. Additionally, the weakened motor function of Myo9 molecules appears to interfere with their actin cross-linking and scaffolding function of the myosin[90,126], which is required for the assembly of bundled actin structures to support maturation and maintenance of stable connections between cells and the underlying matrix[31,35]. We propose that this dual mechanism of interference could be effective in suppressing the metastatic features of cancer cells.

Adhibin displays not only important characteristics of a potential anti-metastatic drug candidate, but its inhibitory potential towards class-IX myosins also provides a central opportunity to expand our current knowledge of this distinct family of motorized signalling proteins in regulating RhoGTPase-dependent mechanisms related to morphogenesis, cancerogenesis, and immune response[33,38].

The identification of a class-specific myosin inhibitor and the exploration of its mechanism of action opens new perspectives in chemical cancer therapies. By specifically improving the properties of the pharmacophore in terms of target affinity, bioavailability, and other pharmacological criteria could be facilitated. Although the selectivity and potency of adhibin provide significant progress towards biologically active small molecule effectors that target Rho-signalling pathways for use in cancer therapy, a limitation of this study is whether adhibin would exhibit its anti-migratory and anti-adhesive properties observed in the various cancer cell models also in the animal tumour model to act as an anti-metastatic drug. In vivo cancer models are necessary to test the efficacy of adhibin on the organismal level, validating class-IX myosins as important therapeutic targets and adhibin as the pharmacological lead in the treatment of metastatic cancers.

## Methods

### Ethics statement
Intestinal biopsies from the transverse colon of a healthy volunteered donor used to generate the human colonoids were collected after informed consent of the donor and institutional review board approval (number 8536_BO_K_2019 from 26.06.2019) of Hannover Medical School (MHH). Animal housing of BALB/cJRj mice and experimental procedures were approved by the animal welfare committee of the Hannover Medical School, complied with the German animal welfare legislation and were finally approved by the Lower Saxony State Office for Consumer Protection and Food Safety (LAVES, AZ 33.12-42502-04-22-00021).

### Reagents
Detailed information about reagents, antibodies, plasmids, kits and oligonucleotides used in this study (catalogue number, RRIDs,

sources) are given in the Supplementary information in the section "Reagent and resource information".

### Cell lines
The cell lines (A549, MLE-12, B16-F1, Caco-2, HeLa, NIH/3T3, and Calu-3), purchased by ATCC, were routinely cultured in Dulbecco's Modified Eagle's Medium (DMEM)-high glucose (Merck) supplemented with 10% fetal bovine serum (FBS, Biowest) and 100 U/mL penicillin, 100 μg/mL streptomycin (Thermo Fisher Scientific) at 37 °C and 5% $CO_2$. All cell lines were routinely checked for Mycoplasma and were kept in culture for a maximum of 3 weeks. Primary macrophages from mice were isolated as described in ref. 127.

### Three-dimensional colonoid cultures
Intestinal biopsies from the transverse colon of a healthy volunteered donor were used for crypt isolation and establishing colonoid cultures according to Sato et al. [128]. Isolated crypts were embedded in 40% Cultrex (BME001, R&D Systems) diluted in stemness medium and supplemented with 1% Jagged-1 peptide (Cat. No. AS-61298, AnaSpec). Three 15 μL domes of hydrogel were plated in each well of a 24-well plate, and covered with 0.5 mL of stemness medium after complete polymerization of the hydrogel. Composition of the stemness medium was L-WRN conditioned medium (prepared as in Miyoshi and Stappenbeck, 2013[96]), Advanced DMEM/F12 (Invitrogen, 12634-028), HEPES 1 M (Invitrogen,15630-056), GlutaMAX-I (Invitrogen, 35050-079), Primocin (Invitrogen, ant-pm-1), N2 supplement (Invitrogen, 17502-048), B27 supplement (Invitrogen, 17504-044), N-Acetylcysteine (Sigma-Aldrich, A9165-5G), human recombinant EGF (Invitrogen, PMG8043), A-83-01 (Tocris, 2939), SB202190 (Tocris, 1264), Nicotinamide (Sigma-Aldrich, N0636) [Leu15]-Gastrin I (Sigma-Aldrich, G9145), CHIR99021 (Sigma-Aldrich, SML1046-5MG), Y-27632 dihydrochloride (Tocris,1254). After 2 days, the stemness medium was replaced with an expansion medium with a similar composition to the stemness medium but without Y-27632 and CHIR-99021. The expansion medium was refreshed every two days. For splitting, colonoids at day 5-6 of culture were mechanically dissociated in a cold medium and fragments were split and cultured as above.

### *Drosophila* stocks maintenance
Oregon R fly line was used in the treatment experiments. Fly stocks were maintained at 25 °C on a standard cornmeal-agar diet in a controlled environment (constant humidity and light-dark cycle). Both male and female flies were studied to avoid limitations of the data generalizability.

### Mice experiments
All mice were housed in a 14/10 h light/dark cycle according to directive 2010/63/EU under specific pathogen-free conditions at the central animal facility of Hannover Medical School and fed with a standard diet (Altromin 1320, Altromin Spezialfutter GmbH & Co. KG, Lage, Germany) and unlimited access to drinking water. BALB/cJRj mice were purchased from Janvier Labs (Le Genest-Saint-Isle, France) and acclimatized for at least two weeks prior to the experimental applications. Twenty-four mice were randomly allocated into four experimental groups each consisting of six mice (three males, three females). Mice were intraperitoneally injected once with 0.142 mg/kg body weight (bw), 0.047 mg/kg bw, 0.016 mg/kg bw (corresponding to serum concentrations of approx. 5.0 μM, 1.67 μM, and 0.56 μM, respectively, assuming 100% resorption), or solvent only (0.5% (v/v) DMSO in PBS). After injection, the mice were clinically monitored for two weeks following a clinical grading system ranging from 1 (no impairment) to 6 (lethally impaired). The gradation takes into account the total body weight, the general state of health (activity, appearance of coat, eyes, and orifice), and the behaviour (attention, posture, movement) of the mice. Finally, mice were subjected to histo-pathological examination

of brain, lungs, heart, liver, spleen, kidney, stomach, intestine, pancreas, and bone marrow applying standard techniques.

## Protein purification

Rabbit fast skeletal muscle heavy meromyosin (*sk*HMM) was prepared as described in ref. 129. All other myosins were purified from native tissue or recombinant from the following organismal sources: *D. discoideum* (*Dd*), *O. coniculus* (*Oc*), *S. scrofa* (*Ss*), *R. norvegicus* (*Rn*), *C. elegans* (*Ce*), *H. sapiens* (*Hs*). Motor domain constructs of myosin-2, myosin-5a–5b, *Ce* myosin-9/*Rn* myosin-9b, were prepared as described previously in refs. 126,130–135. *Hs* nonmuscle myosin-2A (NM2a) was purified from SF9 cells using the baculovirus expression system as described in ref. 130 (for more information see Supplementary Table 5). F-actin was prepared according to the protocol by Lehrer and Kerwar[136].

## In vitro functional assays

Initial testing of the compounds for their inhibitory potency on myosin ATPase activity was performed with skeletal myosin-2 using the malachite green assay[137]. The effect of the compounds on basal and actin-activated $Mg^{2+}$-ATPase activities was measured using the NADH-coupled assay as described previously in ref. 138. Compounds dissolved in DMSO were added to the reaction mixture in the absence of nucleotides and incubated for 20 min before the reaction was started by the addition of ATP. The effect of the compounds on actin filament translocation was analysed with an Olympus IX81 inverted fluorescence microscope as described at 25 °C[139]. Experimental flow cells were constructed using nitrocellulose-coated glass coverslips to which the myosins were immobilized directly via anti-His antibodies. Actin-translocation was initiated with the addition of assay buffer containing 4 mM ATP, 10 mM DTT and 0.5% methylcellulose. 0.5 mg/mL equine cytochrome C was used as a blocking agent. Average sliding velocities were determined from the Gaussian distributions of the translocation distances over time using the automated tracking DiaTrack 3.0 software[140]. Frequency counts and additional data analysis were performed with Origin 8.0.

## Crystallization and X-ray structure determination

*Dd* myosin-2 motor domain construct was co-crystallized with 1 mM compound **5**, 2 mM ADP, 2 mM meta-vanadate and 2 mM magnesium chloride in 0.2 M lithium acetate and 20% PEG 3350 by sitting drop vapour diffusion at 4 °C. The obtained crystals were flash-frozen in a final solution of 0.2 M lithium acetate, 20% PEG 3350, and 20% ethylene glycol as cryoprotectant. Diffraction data was collected at Soleil synchrotron (France) beamline Proxima-2A (wavelength: 0.9794 Å, temperature: 100 K). The data was processed with XDS[141] and scaled with AIMLESS of the ccp4 software suite[142,143]. The *Dd* myosin-2 pre-power stroke structure (PDB-ID: 1VOM[144]) was used as a search model for molecular replacement using Phaser[145]. Model building and structure refinement were carried out using Coot[146] and Phenix.refine[147]. Ligand restraints for the compound were generated with eLBOW[148]. The final model and the structure factor amplitudes have been deposited in the Protein Data Bank (www.rcsb.org)[149] with PDB accession code: 6Z2S (Ramachandran statistics: favoured: 97%, allowed: 3%, and outliers: 0). Refinement statistics are listed in Supplementary Table 2.

## Homology modelling and molecular docking

Homology models of the myosin-9a and myosin-9b motor domains were generated using Modeller 9.16[150] and the obtained co-crystal structure of *Dd* myosin-2 in complex with **5** (pdb: 6z2s) as the template. Twenty structural models were generated and the Modeller objective function and the discrete optimized protein energy (DOPE) score were used for evaluating and selecting the best model. As no suitable template was found for the extended loop 2, this loop was neglected. Molecular docking was performed with Autodock4

and the Lamarckian Genetic Algorithm[151]. Proteins and ligands were preprocessed using Schrödinger Maestro, and energy-minimized with MacroModel Release 2019-3 and the OPLS3 force field[152]. Energy minimization was performed with the Polak-Ribiere conjugate gradient algorithm to a gradient of 0.0001 kJ mol$^{-1}$ Å$^{-1}$. Subsequently, the structures were processed with AutodockTools. Thirty docking runs were carried out per small molecule and protein with an initial population size of 150 and a maximal number of evaluations of 2.5 billion. Convergence of the docking experiments and selection of the best poses were done using cluster analysis and the predicted binding energies. The homology data are provided as Supplementary Data 1, with a description in the 'Description of Supplementary Data 1' file.

## Microscale thermophoresis (MST)

The binding affinities of **12** and **5** to myosins were measured using the Monolith NT.115 Microscale Thermophoresis device[50,153]. Myosin motor domain constructs were labelled with an atto-647 maleimide dye in 20 mM HEPES pH 7.3 mM and 100 mM NaCl for 30 min at 25 °C. MST measurements were performed at 25 °C using a myosin concentration of 10 nM in an experimental buffer containing 20 mM HEPES pH 7.3, 100 mM NaCl, 0.5 mg/mL BSA and 0.05% Tween-20. Prior to the measurements, the compounds were added to the samples at concentrations of 50 nM to 400 μM. After 15 min incubations, samples were centrifuged at 15,000×*g* for 5 min at 4 °C and immediately measured. Inhibitor-binding affinities were obtained using MO.

## qPCR

Expression of *MYO9A* and *MYO9B* was determined by quantitative real-time PCR (qPCR). Total RNA extracts were isolated using the RNeasy Mini Kit (Qiagen) according to the manufacturer's instructions. RNA quality was assessed by agarose gel electrophoresis. Reverse transcription was performed with the QuantiNova Reverse Transcription Kit (Qiagen) using 1–5 μg RNA according to the manufacturer's protocol with slight modifications. Removal of genomic DNA was performed for 5 min at 45 °C. For reverse transcription, an initial primer annealing time of 5 min at 25 °C was chosen followed by a 30 min elongation period at 45 °C and heat inactivation for 10 min at 85 °C. PCR was carried out in a StepOne-Plus Real-Time PCR system (Applied Biosystems, Thermo Fisher Scientific) using 5 μL cDNA (diluted 1:100 to 1:500). The qPCR reaction mixture contained 1× Power SYBR Green Master Mix (Thermo Fisher Scientific) and 0.25 μM of forward and reverse primer in a total volume of 14 μL. Initial denaturation was performed for 10 min at 95 °C. PCR was performed over 40 cycles, each comprising a 15 s denaturation step at 95 °C and primer annealing and elongation for 1 min at 60 °C. A final melting curve was generated to evaluate primer specificity. $C_t$ values were calculated with the StepOne software version 2.1 with a cycle threshold of 0.2. Quantification of myosin-9b expression was performed with the $2^{-\Delta\Delta Ct}$-method using the peptidylprolyl isomerase A (*PPIA*) as a housekeeping gene. cDNA samples were analysed in duplicates.

## Drug solubility

Prior to each experiment, stock solutions of the compounds dissolved in 100% DMSO, were further diluted in DMEM supplemented with 1% FBS (FBS, Biowest) to obtain a 1% DMSO (*v/v*) solution with the respective drug concentration. The samples were kept at 37 °C degrees for 1 h and then centrifuged at 20,000 × *g* for 20 min. Supernatants and remnants re-dissolved in 120 μL DMSO, were transferred to 96-well plates and the relative changes in maximum absorption were measured spectrophotometrically (Microplate Reader, BioTek Synergy 4) to determine solubility.

## Cell transfection and drug administration

The addition of adhibin was always done in DMEM-high glucose supplemented with 1% FBS, to avoid the formation of complexes between albumin and adhibin. The drug was prediluted in 100% DMSO and prior to each experiment further diluted in DMEM to obtain a 1% DMSO solution ($v/v$) with the respective drug concentration. When needed, the plates were pre-coated with 25 µg/mL laminin (Sigma), 20 µg/mL fibronectin (Sigma), or 1 µg/mL Poly-L-lysine (Sigma) for 1 h at room temperature, followed by two washes with 1× PBS. The cells were transfected using the jetPRIME transfection reagent (Polyplus) following the instructions of the manufacturer. For A549 cells 2 µg of DNA was used, whereas for B16-F1 cells were transfected with 1 µg DNA. Experiments were performed at least 24 h after transfection. Plasmid dtomato-2xrRBD (129625, Addgene) was cloned into the mammalian pEGFC1 vector using *AgeI/XbaI* restriction sites to create pEGFC1-dtomato-2xrRBD. β-actin was cloned from the construct described in refs. [154], using *EcoRI/XbaI* restriction sites to create pEGFC2-β-actin. pEGFPN1-Paxilin and mScarletN1-LifeAct were a kind gift from Jan Faix (Hannover Medical School, Germany), pEGFPC1-Myr5 was described previously in ref. [27]. siRNAs (#3, #4, and #5) were obtained from QIAGEN with the reference numbers Hs_MYO9B_3 (SI00653709), Hs_MYO9B_4 (SI00653716), Hs_MYO9B_5 (SI03125661), and the scrambled RNA scRNA (4390847).

## Cytotoxicity and cell growth

Compound toxicity was analysed using the neutral red uptake assay as described previously in ref. [155]. Shortly, 17.500 A549, 20.000 MLE-12 or 30.000 B16-F1 were seeded in 96 multi-well plates (Nunc, Thermo Fisher Scientific) and allowed to grow for 18 hr. Inhibitors were added to cells after dilution in a cell culture medium containing 1% FBS. After 18 h the cell culture medium was replaced by fresh medium containing 1% FBS and neutral red (40 µg/mL, Sigma-Aldrich). After a 2 h incubation and subsequent washing with PBS (140.0 mM NaCl, 2.7 mM KCl, 10.0 mM Na$_2$HPO$_4$, 1.8 mM KH$_2$HPO$_4$, pH 7.4), 150 µL destaining solution (48% ethanol, 1% acetic acid in distilled water) was added per well. The absorption in each well was measured at 540 nm using a SPECTROstar Omega multi-plate reader (BMG Labtech). Absorption was normalized to the value obtained for 1% DMSO-treated cells that served as control. To obtain growth curves, a total of approx. 50.000 A549 or MLE-12 cells were seeded and incubated in DMEM supplemented with 1% FBS for 6 h. Then, cells were washed twice with PBS and further grown in 1% FBS DMEM containing 1% DMSO, 10 µM adhibin, or 25 µM adhibin for up to 48 h. For B16-F1 cells, approximately 50.000 cells were seeded directly in 1% FBS DMEM with 1% DMSO, 5 µM adhibin, or 10 µM adhibin. At defined time points, non-adherent cells were harvested by centrifugation ($1000 \times g$, 10 min) and resuspended in 30 µL of medium. Adherent cells were trypsinized, pelleted by centrifugation, and resuspended in a 30 µL medium. Cell suspensions were mixed with trypan blue (Sigma-Aldrich) at a 1:1 ratio and the number of viable and dead cells was determined using a Neubauer chamber.

## Cell viability, cell attachment and cell detachment assays

To assess the time-dependency of attachment of A549 and MLE-12 cells in the presence of different myosin inhibitors, cells were trypsinized, centrifuged, and resuspended in 1% FBS DMEM. An equal volume of the cell suspension was added to each well of a 96-well plate before the addition of the myosin inhibitors or 1% DMSO. The cells were allowed to attach for 0.5 h, 1 h, 2 h, 4 h, or 6 h in the incubator before the medium was removed and floating cells were removed. A neutral red-containing medium was added to the cells for 2 h and the subsequent assessment of cell numbers was performed as described above for the neutral red uptake assay. The relative number of attached cells was calculated by normalisation of the absorption values to those of cells

after 6 h. To assess the effect of the inhibitors on cell adhesion, cells were trypsinized, washed and mixed with cell culture medium with or without inhibitor and placed in 24-well plates (Nunc, Thermo Fisher Scientific). The cells were allowed to adhere for 6 h before the supernatant was collected. Viable and dead cells both in the supernatant and adhered were counted using a Neubauer chamber after mixing them 1:1 with trypan blue (Sigma-Aldrich).

## Cell migration and cell spreading

Collective cell migration was investigated using a scratch assay. Shortly, cells were grown to confluence in 96-well µ plates (Ibidi). After scratching the plate surface with a pipette tip, the cell culture medium was exchanged twice before a solution of the compound in DMEM medium containing 1% FBS and 1% DMSO was added to the cells. After defined time points, scratch closure was visualized using an inverse Eclipse TS1000 brightfield microscope (Nikon). To assess random migration, approx. 15.000 cells were seeded into each well of a µ-slide 8-well glass bottom dish (Ibidi) pre-coated with laminin in sterile filtered DMEM with 10% FBS and left to adhere for 1 h. 30 min before the acquisition, the medium was changed to either DMEM with 1% FBS, or DMEM with 1% FBS and 1% DMSO, 5 µM of adhibin or 10 µM of adhibin, supplemented with 25 mM HEPES (Sigma). Movies of migration were recorded with a Nikon microscope with an Andor/Yokogava Spinning disk equipped with a 10X lens with a frame rate of 60 s for 3 h. Radar plots were created with SigmaPlot (Systat Software GmbH). To measure cell spreading, approx. 20.000 B16-F1 cells were seeded in each well of an µ-slide 8-well glass bottom dish (Ibidi) pre-coated with laminin in sterile-filtered DMEM with 1% FBS, or DMEM with 1% FBS and 1% DMSO or 5 µM of adhibin, always supplemented with 25 mM HEPES (Sigma). The plate was transferred immediately to an Olympus IX-83 Inverted microscope, and images were taken with a 10x objective at frame rates of 1 min.

## RhoA activation assay

RhoA activity was measured with the G-LISA RhoA Activation Assay Biochem Kit according to the manufacturer's instructions (Cytoskeleton: BK124, shown in Fig. S4R). For sample preparation, $2.5 \times 10^6$ cells were seeded in a T175 cell culture flask and incubated in 5% CO$_2$ at 37 °C overnight. After removal of the medium, cells were washed twice with PBS and incubated in fresh DMEM medium supplemented with 1% FBS containing 1% DMSO (control) or 1% DMSO and 25 µM adhibin for 6 h. After removal of the medium, prechilled Kit-lysis buffer (100 µL) containing a freshly added protease inhibitor cocktail was added to the cells and cells were collected with a scraper. Lysates were centrifuged at $10,000 \times g$ for 1 min at 4 °C and the total protein concentration of the supernatants was determined spectrophotometrically using the Precision Red Protein Assay Reagent provided in the kit. Concentrations were adjusted to 2.0–2.5 mg/mL, flash-frozen in liquid nitrogen, and stored at −80 °C. All following procedures were conducted according to the manufacturer's instructions using a BioTek plate reader for absorbance measurements. The measurements were then normalised to control and are shown as relative percentages. Additionally, the RhoA Pull-Down Activation Assay Biochem Kit (Bead Pull-Down Format, Cytoskeleton: BK036, shown in Fig. S4Q) was used according to the manufacturer's instructions to measure active RhoA. For the immunoblotting procedure, first, the lysates were matched using an SDS page. Equal amounts of lysates were used. Blocking was performed with 10% FBS in TBST for 1 h, followed by incubation with anti-RhoA diluted in 10% FBS in TBST (1:100, Santa Cruz Biotechnology) overnight at 4 °C. The membranes were incubated in phosphatase-conjugated anti-mouse antibody for 2 h at room temperature and were developed with 20 mg/mL of 5-brom-4-chlor-3-indolylphosphat-p-toluidin (BCIP) in NaHCO3, pH 10.0. The rest of the procedure is described in the section "Immunoblotting".

## Immunofluorescence microscopy

For immunofluorescence experiments, cells were cultured on glass cover-slips (for B16-F1 cells coated with laminin) in 24-well plates (Nunc, Thermo Fisher Scientific) in DMEM containing 1% FBS in the absence or presence of the compounds overnight. B16-F1 cells were fixed with 37 °C-prewarmed 4% paraformaldehyde in PBS for 20 min at 37 °C. The rest of the cells were fixed with 4% paraformaldehyde in PBS (Santa Cruz Biotechnology) for 20 min at 4 °C. After washing with PBS supplemented with 0.1 M Glycin and 0.02% sodium azide, cells were permeabilised with 0.1% triton X-100 (Merck) in PBS for 3 min at room temperature (for A549 and MLE-12 cells) or with 0.05% triton X-100 (Merck) in PBS for 30 s at room temperature (for B16-F1 cells). Blocking was performed with 5% bovine serum albumin (BSA, Carl Roth) in PBS for 1 h at room temperature. The following primary antibodies were used at dilutions as defined: anti-Myo9a (1:500, Invitrogen), anti-Myo9b (1:1000, Proteintech), anti-paxillin (1:100, BD Biosciences), anti-ZO1 (1:300, Thermo Fisher Scientific), anti-pMLC2 (1:200, Cell Signalling), anti-NMIIA (1:200, Biolegend), anti-NMIIB (1:500, Biolegend), anti-vinculin (1:200, Sigma-Aldrich), anti-VASP (1:1000, form Jan Faix described in ref. [65]). All primary antibodies were diluted in PBS containing 2.5% BSA. Incubations of cells with primary antibodies were performed overnight at 4 °C or for 2 h at 37 °C. AlexaFluor555-conjugated secondary anti-rabbit (1:300, Thermo Fisher Scientific) and AlexaFluor488-conjugated anti-mouse (1:300, Thermo Fisher Scientific) antibodies were added to the cells together with AlexaFluor633-conjugated (1:300, Thermo Fisher Scientific) or Atto633-conjugated phalloidin (1:300, Atto-tec) in 2.5% BSA in PBS for 1 h at room temperature. The cells were mounted in Prolong Gold antifade mountant with DAPI (Invitrogen) and stored at 4 °C. The cells were imaged using a ZEISS 980 with an Airyscan 2 microscope equipped with a 40× or a 63× oil immersion objective. For TIRF microscopy to visualize FA at A549 cells, cells were first pre-extracted in 0.3% PBS-Triton and 2% PFA in PBS for 2 min at room temperature, followed by fixation and staining as described. The cells were imaged with an Olympus IX-83 Inverted microscope, using a 60x oil objective and a 488 nm TIRF laser.

## Live cell imaging

To visualize actin dynamics, A549 and MLE-12 cells were plated in glass bottom dishes (ibidi), left overnight to attach and then imaged in 1% FBS DMEM in the presence or absence of the compound supplemented with 25 mM Hepes. B16-F1 cells were transfected with mScarlet-LifeAct, plated in laminin-coated glass bottom dishes (ibidi) and imaged 1 h after seeding in 1% FBS DMEM in the presence or absence of the compound, supplemented with 25 mM Hepes. The acquisition was performed using an Olympus IX-83 Inverted microscope equipped with a 60× oil objective. Focal adhesions (FA) were visualized in A549 cells transfected with pEGFP-paxillin with an Olympus IX-83 Inverted microscope, using a 60× oil objective and a 488 nm TIRF laser. The cells were seeded on glass bottom dishes and left overnight to attach. Cells were transfected as described. Movies were taken 24 h after transfection in DMEM containing 1% FBS and 1% DMSO or 25 μM adhibin supplemented with 25 mM HEPES at frame rates of 1 min for up to 10 h. To visualize activated RhoA (RhoGTP), B16-F1 were transfected with dtomato-rothekin as described in ref. [68]. Cells were seeded in laminin-coated glass bottom dishes and imaged 1 h after seeding in 1% FBS DMEM in the presence or absence of the compound, supplemented with 25 mM Hepes, using a 60× oil objective of the Olympus IX-83 Inverted microscope. The image acquisition interval was 5 s. FRAP experiments were performed using a ZEISS 980 with an Airyscan 2 microscope equipped with a 63× oil immersion objective. Bleaching was performed with a 488 nm diode laser for 0.5 ms. The imaging interval was 2 s.

## BrDU assay

Before fixation, cells were prepared as described in the section 'Immunofluorescence'. 24 h prior to fixation the cells were treated with DMEM 1% FBS with or without the presence of 1% DMSO, 5 μM, 10 μM, or 25 μM adhibin, all supplemented with 10 μM of BrDU in PBS. The cells were then fixed with 4% PFA for 15 min at room temperature, permeabilized with 0.1% Triton in PBS for 20 min at room temperature and subsequently treated with 1 N HCl for 10 min on ice, followed by 2 N HCl for 10 min at room temperature. Then, the HCl was neutralized with an equal volume of 0.2 M Na$_2$HPO$_4$ and 0.1 M citric acid (pH = 7.4) for 10 min at room temperature, followed by three washes with 0.1% triton in PBS. Primary antibody: BrDU monoclonal antibody (1:200, MA3-071, Invitrogen) in 0.03% Triton and 2.5% BSA in PBS, for 2 h at room temperature. The following secondary antibodies were used at dilutions as defined: AlexaFluor647-conjugated goat anti-mouse antibody (1:300, A21235, Thermo Fisher Scientific) in 0.03% Triton and 2.5% BSA in PBS, for 1 h at room temperature. The cells were mounted in Prolong Gold antifade mountant with DAPI (P36935, Invitrogen) and stored at 4 °C.

## Fluorescence-activated cell sorting (FACS)

For FACS experiments, cells were fixed and stained in solution with Atto488 phalloidin (0.5 μM, Atto-tec) to quantify F-actin and DAPI (1:500, 1 mg/mL, Sigma) to perform cell cycle analysis as described in the section Immunofluorescence. Then, the cells were sorted using a FACSAria III Fusion. Analysis was performed FlowJo™ v10.8 Software (BD Life Sciences). Detailed FACS data and panels are included in the Supplementary information.

## Immunoblotting

For Immunoblotting experiments, cells were cultured in 6-well plates (Nunc, Thermo Fisher Scientific) until 90% confluency in DMEM containing 1% FBS in the absence or presence of the compound for 18–24 h. The cells were then washed twice with ice-cold PBS, scraped from the dishes, and centrifuged for 1.5 min at 300×$g$ in a tabletop centrifuge. Then, the supernatant was discarded and the pellet was resuspended in 200 μL RIPA Buffer (150 mM NaCl, 1% Triton x-100, 0.1% SDS, 50 mM Tris, 5 mM DTT, 5 mM MgCl2, pH = 8 at 4 °C), supplemented with 5 mM benzamidine, Mix1 inhibitors (500 mg Na-p-tosyl-L-arginine-methyl-ester-hydrochloride, 400 mg Tosyl-phenylalanyl-chloromethyl-ketone, 10 mg pepstatin, 2 5 mg leupeptin, diluted in 50 mL ethanol) 1:100, Mix2 inhibitors (100 mM Phenylmethylsulfonylflourid in ethanol) 1:100, benzonase 1:10000, and 5 mM of ATP, and was kept for 1 h at constant agitation (wheel rotator) at 4 °C. Cell remnants were then spun down for 25 min at 4 °C. The protein concentration was determined by a Bradford assay using BSA solutions of defined concentrations in the same RIPA buffer. Absorption measurements were done with a Synergy 4 fluorescence microplate reader (Biotek, Bad Friedrichshall, Germany) at 595 nm excitation wavelength. For immunoblotting, 30 μg total protein from A549 cell lysates and 50 μg total protein from MLE-12 and B16-F1 cell lysates were loaded onto an acrylamide gel. The supernatant was mixed with 1:5 volume of 5x Laemmli Buffer 10% SDS, 100 mM DTT, 50% glycerol, 250 mM Tris pH 6.8, and 0.025% of bromophenol blue, snap frozen with liquid nitrogen and stored at −80 °C until use. To prevent dephosphorylation, the lysates for pMLC2 and pMYPT1 blots were made by scraping the cells from the plate directly in 1.5x Laemmli Buffer, 10% SDS, 100 mM DTT, 50% glycerol, 250 mM Tris pH 6.8, and 0.025% of bromophenol blue, at 4 °C. The cell lysates were boiled for 5 min at 95 °C before loading. They were loaded into BioRad 4−20% Mini-PROTEAN TGX Precast Protein Gels (BioRad) or homemade gels (see Source Data) and run in 1x Laemmli buffer with 0.1% SDS for 90 min at 100 V. The buffer used for the transfer of the proteins to the nitrocellulose membranes via semidry blotting was composed of

25 mM TrisHCl, 192 mM Glycin and 20% Methanol. 1x TBSTween (50 mM Tris, 150 mM NaCl, 0,05% Tween) with 5% dry milk was used for blocking for 45 min. Antibodies were diluted in 1x TBST with 5% dry milk. The following primary antibodies were used at dilutions as defined: anti-Myo9b (1:500, Proteintech), anti-Myo9a (1:500, Invitrogen), anti-pMLC2 (1:500, Cell Signalling), pan anti-actin (1:1000, Abcam), anti-NMIIA (1:500, Biolegend), anti-NMIIB (1:500, Biolegend), anti-pMYPT1 (T850) (1:500, Upstate/Sigma-Aldrich), anti-vinculin (1:500, Sigma-Aldrich), anti-VASP (1:1000, form Jan Faix described in ref. [65]) anti-GAPDH (1:10000, MERCK), all overnight at 4 °C at constant agitation. The following secondary antibodies were used at dilutions as defined: Stabilized peroxidase-conjugated goat anti-rabbit (H + L), HRP conjugated (1:500, Invitrogen), goat anti-mouse (H + L), HRP conjugated (1:500, Invitrogen), all for 45 min at room temperature and constant agitation. For the determination of G-actin and F-actin, cells were grown to approx. 90% confluence in 10% FBS containing DMEM in 10 cm plates. The medium was changed to DMEM supplemented with 1% FBS and 1% DMSO or DMEM supplemented with 1% FBS and compound (10 μM for B16-F1 cells and 25 μM for A549 and MLE-12 cells) for 18 h. The G- and F-actin ratios were determined for the entire cell collective including both, adherent and non-adherent cells, which detached from the surface due to drug treatment. The non-adherent fraction was collected by centrifugation at 1000×$g$ for 1 min. Adherent and non-adherent cell fractions were washed twice with ice-cold PBS and combined in a total volume of 400 μL ice-cold lysis buffer (25 mM HEPES, pH = 7.2, 100 mM NaCl, 10 mM Na-P, 3% sucrose, 2 mM MgCl$_2$, 10 mM KCl, 5 mM β-mercaptoethanol, 5 mM EGTA, 5 mM ATP, 0,1 mM AEBSF, 5 mM Benzamidine, 0.5% NP-40, 0.5% Triton X-100, 1 μM phalloidin) and lysed on ice for 15 min under shaking. 200 μL of the lysates were centrifuged for 1.5 h at 150,000 × $g$ at cold to isolate phalloidin-stabilized F-actin, while G-actin remained in the supernatant. The supernatant was mixed with 100 μL 3× SDS buffer and the pellet with 300 μL 1× SDS buffer containing 50 mM NaCl to prevent depolymerization of F-actin. Both fractions were applied to SDS page electrophoresis and the amount of actin was determined by immunoblotting. The G-to-F-actin ratio was calculated from the relative intensities between supernatant (G-actin) and pellet (F-actin). GAPDH quantification served as a reference. Analysis of intensity profiles was performed with Fiji[156].

## Spheroid formation and 3D migration

To study cellular 3D migration, we used spheroid models, which mimic the three-dimensional composition of tissue and micro-tumors. The spheroids we obtained by seeding NIH 3T3 mouse fibroblasts (4000 cells/mL) in a 96-well plate coated with 1% agarose (sterile) in 10% DMEM that was grown for 3 days at 37 °C, 5% CO$_2$ in full isolation. Matured spheroids were washed once in PBS and transferred into a 30 μl drop of Matrigel (356264, Corning) on an 8-well glass bottom dish. After polymerization (20 min at 37 °C), the Matrigel was covered with 1% DMEM in the absence and presence of adhibin. This time point was marked as 0 h. Spheroids were imaged at time points 0 h, 16 h, 24 h, 32 h, 48 h, 60 h and 72 h using an Olympus IX-83 Inverted microscope equipped with a 10× air objective. The 3D migration ability of the cell collective was related to the change in spheroid perimeter per time. To analyse the influence of adhibin on the ability of cells to form spheroids through 2D and 3D migration, 4000 NIH 3T3 fibroblasts/mL were seeded in 96 well plates coated with 1% agarose (sterile) in 1% DMEM in the presence and absence of the drug following the same procedure and conditions described above. After 72 h, the status of spheroid formation was imaged using an Olympus IX-83 Inverted microscope equipped with a 10x air objective. The perimeter of the spheroids was used to quantify the progress of the 3D assembly of the cells.

## Trans epithelial electrical resistance (TEER) assay

The barrier function was analysed by measuring the transepithelial electrical resistance (TEER) with the cellZscope (nanoAnalytics, Muenster, Germany). For the measurement, $3 \times 10^4$ Caco-2 cells per transwell insert were seeded and cultured with the cell culture medium for two days before being placed in the cellZscope. The transwell inserts have a transparent PET membrane with a pore size of 0.4 μM (BD Falcon, Corning). After the establishment of a barrier of at least 700 Ωcm$^2$ (after 5 days), cells were treated with adhibin (2 μM, 10 μM, 25 μM, or 40 μM) or 0.3% DMSO (control) and TEER was monitored every hour for 48 h. The TEER data were automatically recorded by the cellZscope software (nanoAnalytics). The cells used for the TEER measurements were used for staining tight junction proteins. The cells were washed with PBS before fixation with an ice-cold acetone/methanol mixture (50:50) for 5 min at −20 °C. Cells were blocked with 1% bovine serum albumin (BSA, Roth) for 1 h at 37 °C after washing with PBS. The primary antibody anti-ZO-1 (1.25 μg/mL Thermo Fisher Scientific, 40-2200) in PBS was added to the cells overnight at 4 °C. The cells were washed with PBS before secondary antibodies were added for 1 h at 37 °C. The secondary antibody iFlour™488 anti-rabbit (ATT Bioquest, 16608) was diluted 1:500 in PBS. After washing with PBS, the transparent PET membranes with the cells were excised placed on a glass cover-slip and mounted in Mowiol (Roth) at 4 °C. Cells were analysed by confocal laser scanning microscopy using a Nikon TM200 inverse confocal laser scanning microscope (Nikon, Düsseldorf, Germany) equipped with a 60× water immersion objective. Eight to ten images per transparent PET membrane were taken.

## Colonoid drug treatment

To evaluate the function of adhibin on colonoid cultures, Y-27632 was omitted from the stemness medium in the experimental set. Media with different concentrations of adhibin, as well as the DMSO control contained 0.1% vol/vol final concentration of DMSO. Samples labelled with "untreated" received neither DMSO nor adhibin. During optimization experiments, treatment with DMSO/adhibin was carried out immediately after splitting of the colonoids. Later this was postponed to day 2, to avoid interfering with the closure of the colonoid fragments into spheroid structures by Adhibin. Cultures were maintained in an expansion medium for up to 8 days and imaged every day using a Zeiss Axio microscope with a 5× objective.

## Drosophila immunohistochemistry

Flies were kept either on a standard cornmeal-agar diet with diluted DMSO or 50 μM adhibin. Fly brains were dissected in 1× phosphate buffered saline (1× PBS) and then fixed in 4% formaldehyde diluted in 1× PBS for 20 min at room temperature. Next, brains were washed with PBT (0.2% Triton X-100 in 1× PBS) 4 times, followed by a block with PBTB (2 g/L bovine serum albumin, 5% normal goat serum, 0.5 g/L sodium azide) for 1 h at room temperature and then incubated at 4 °C in with primary antibodies diluted in PBTB on nutator overnight. The following day, samples were washed with 1x PBT four times followed by block for 1 h and 2 h incubation with secondary antibodies at room temperature. Next, samples were washed 4 times with PBT (one of the washes contained DAPI to mark nuclei). Lastly, medium (70% glycerol, 3% n-propyl gallate in 1× PBS) was added to samples for later mounting on the slides. For embryo staining flies were allowed to lay eggs at agar plates with yeast overnight. Eggs from agar plates were collected and treated with 50% bleach for 2 min to remove chorion. Then embryos were washed 3 times with water. Embryos were fixed in the solution containing 1:1 heptane and 4% PF in PBS for 20 min. After fixation, the lower part containing the fix was removed and an equal amount of methanol was added. Then embryos were vigorously vortexed for 2 min to remove the extraembryonic membrane. Next, the upper phase continuing heptane and embryos that did not sink were removed and fresh methanol was added. To rehydrate embryos were

washed 3 times with PBT followed by 1 h block in PBTB and finally primary antibody incubation overnight. The following day the same protocol procedure as for fly brain immunohistochemistry was followed. The following primary antibodies were used: mouse anti-24B10 (1:50), mouse anti-CoraC (1:50), mouse anti-FasII (1:50), rat anti-DE-Cadherin (1:50) from the Developmental Studies Hybridoma Bank (DSHB); rabbit anti-NrxIV (1:1000 from Christian Klämbt); rabbit anti-HRP (1:1000, Rockland). The following secondary antibodies were used: goat anti-rat Alexa 488 (1:500, Thermo Fisher Scientific), goat anti-rabbit Alexa 488 (1:500), and goat anti-mouse IgG1 Cy3 (1:500, Jackson ImmunoResearch Laboratory). For visualization of cell nuclei, DAPI dye was used (1:1000, Sigma). Samples were analysed using a confocal microscope (Zeiss LSM 700). For making figures, Adobe Photoshop software was used.

## Liquid chromatography coupled mass spectrometry

For the determination of the adhibin concentration in *D. melanogaster*, adult flies and fly larvae were used. Therefore, 25 flies, respectively 10 larvae were transferred to a 2.0 mL tube filled with garnet matrix and a ¼ inch ceramic sphere, 800 µL extraction solvent (acetonitrile/methanol/water 2/2/1) was added and the samples were homogenized in a FastPrep-24® system (MP Biomedicals, Germany device). The samples were subsequently centrifuged and the supernatant was evaporated to dryness at 40 °C under a nitrogen stream. The pellet was dissolved in sample solvent (MeOH/H$_2$O 50/50 containing 0.06 µM efavirenz as Internal standard). The adhibin content in mouse serum was determined as follows: 30 µL of serum was treated with120 µL of ice-cold extraction solvent (acetonitrile/methanol/ water 2/2/1) containing 0.05 µM efavirenz as internal standard. After mixing, the samples were frozen at −20 °C overnight to complete protein precipitation. Subsequently, the samples were thawed and centrifuged or 10 min at 20,800×$g$ (at 4 °C). The supernatant was evaporated to dryness at 40 °C under a nitrogen stream and the pellet was dissolved in 100 µL methanol/water. Adhibin was analysed by liquid chromatography coupled with mass spectrometry. The reversed-phase chromatography was performed on a Shimadzu HPLC system (Shimadzu, Duisburg, Germany) consisting of two HPLC pumps (LC-10AD vp), a temperature-controlled autosampler (SIL-HTc), a degasser (DGU-14A) and a column oven (CTO-10AS). The chromatography column used was a ZORBAX Eclipse XDB-C18 1.8 µ 50 × 4.6 mm (Agilent Technologies, Santa Clara, California, USA) with a Security Guard (C18 (Phenomenex, Aschaffenburg, Germany). The column temperature was kept constant at 30 °C. The flow rate was 0.4 mL/min. The injection volume was 20 µL. Chromatographic separation was performed using a linear gradient. To separate polar sample components, the column was rinsed with 98% eluent A (3/97 methanol/water; 50 mM ammonia acetate; 0.1% acetic acid) for 3 min after sample application. This was followed by a linear gradient whereby the amount of eluent B (97/3 methanol/water; 50 mM ammonia acetate; 0.1% acetic acid) was increased to 95% within 9 min. This concentration was kept constant for 5 min. The column was then re-equilibrated for 6 min. Under these conditions, the retention time of the adhibin was 5.5 min. The detection and quantification of adhibin were performed on a Tandem mass spectrometer (API4000™; Sciex, Framingham, Massachusetts) in positive electrospray ionization mode. Adhibin showed a prominent signal at a mass transition of $m/z$: 353.7 → 323.7. The LC and mass spectrometer were controlled and data were recorded using Analyst software (version 1.7, Sciex, Framingham, Massachusetts). For the quantification of adhibin, a calibration curve (*D. melanogaster:* 0.00364 pmol/sample−75 pmol/sample; mouse serum: 9.77–5000 nM) was created by plotting the ratios of the peak areas of adhibin and the internal standard against the nominal concentration. The calibration curve was calculated using quadratic regression and 1/× weighting.

## Quantification and statistical analysis
### Analysis and quantification of data
**Fluorescence intensity and size.** Fluorescence intensities were quantified as the mean of six randomly chosen regions (10 × 10 pixels) after subtraction of the average background intensity from six randomly chosen regions (10 × 10 pixels) outside the cell. Fluorescence intensities at lamellipodia were quantified as the mean of six random line scans (excluding microspikes) after subtraction of the background intensity determined from the average of six random scan lines outside the cell. The width of lamellipodia was obtained as the average of six edge-to-edge distance measurements. The length of microspikes was determined as the average length of all microspikes per cell. Area and height were measured with Fiji[156]. For height, first, a reslice of the *z*-stack containing the whole cell was created with the Reslice function of Fiji and then the height was measured as the distance from the bottom to the maximum top of the cell.

**Cell migration and spreading.** The cell-free scratch area in the collective migration experiments was calculated using the Scratch Wound Assay plugin[157] of FIJI. For analysis of random cell migration, individual cells were tracked manually using Fiji´s MTRACKJ plugin[158] and movement was analysed using the macro Excel file described in Litschko et. al., to extract speed and directionality parameters[159]. Cell spreading was generally completed after 15–20 min, when control cells started migrating. This time window was used to calculate the cell area of spreading manually using Fiji.

**FA quantifications.** The number and area of FAs were measured from the paxillin channel with a Fiji macro described by Hein et al. [160]. Colocalization of Myo9b and paxillin was analysed by Pearson's coefficient analysis by applying the Fiji plugin JACoP[161] to individual FAs marked by a bounding square. FA dynamics were quantified using the autocorrelation analysis as described in ref. 162, applying the Otsu thresholding method with Fiji. The mean intensity values over time were normalized to the intensity in the first frame to obtain the autocorrelation curve. Additionally, the sliding speed of individual focal adhesions was tracked in each cell using the Trackmate plugin in Fiji[163]. The lifetime threshold of single particles was a minimum of nine frames.

**FRAP analysis.** Analysis of fluorescence recovery after photobleaching was performed with the Zeiss ZEN 3.8 software.

**TEER experiments.** For image analysis of the TEER experiments, the background was first smoothed with the rolling ball method from Fiji. Further analysis of the tight junction architecture was done with the macro TiJOR parameter quantification of the tight junction as described in ref. 164. In our case, ten polygon iterations, with a step size of 33 pixels were applied. The average of 15 experiments (three cultivation cycles) is shown.

**Colonoid quantifications.** The size, shape and survival of the colonoids were analysed with Fiji. The proportional growth of colonoids for each condition at days 4, 6, and 8 was calculated by measuring the cross-section area of single colonoids over time and normalizing them to the respective value of day 2 as the starting point of treatments. Eccentricity was defined as 1/circularity, which was directly derived from measurements with Fiji. For quantitative measurement of Ki-67, MYO9a and MYO9b mRNA expressions, total RNA of day 6 cultures with different treatments for 4 days was isolated and analysed by RT-qPCR.

***Drosophila* quantifications.** To measure the average body mass of adult females and males, the mass of approx. 100 females or 50 males of control and adhibin-treated flies were measured and divided by the

number of flies. The body size of ten adult females and ten adult males treated with DMSO or 50 μM adhibin was measured by a Zeiss AXIO Zoom. V.16. Microscope. Student *t*-test was used to measure the statistical significance.

**Graph and figures preparation.** Preparation of graphs was performed with GraphPad Prism 8 and Origin 2017. Figures were prepared with Microsoft PowerPoint and Inkscape Project (2020).

## Statistical analysis

Data were tested with GraphPad Prism for normal distribution with D'Agostino & Pearson test. Then unpaired, two-tailed *t*-tests (for two-column comparison) and ANOVA (for more than two-column comparison) were performed, either parametric, if the data followed a normal distribution, or nonparametric if the data did not follow a normal distribution. In all plots, centre lines represent mean values, and whiskers show standard deviations (s.d.), unless otherwise indicated. In dot plots, each dot represents a measurement from one cell unless indicated otherwise. *p*-values in all graphs are: $^*<0.01$, $^{**}<0.001$ and $^{***}<0.0001$, $^{****}<0.00001$, n.s. not statistically significant and *n* is the number of cells analysed, unless indicated otherwise in the corresponding figure legend.

## Reporting summary

Further information on research design is available in the Nature Portfolio Reporting Summary linked to this article.

## Data availability

Source data are available with this manuscript as a Source Data file. Source data for each figure are provided. Uncropped versions of all Western Blots are found in the Source Data file. The PBD file of the crystal structure generated in this study has been deposited at PBD under the accession code: 6Z2S. All compounds and plasmids generated in this study are available upon request. Source data are provided with this paper.

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

## Acknowledgements

We thank the beamline scientists of Proxima-2 (Soleil synchrotron) for excellent support during data collection, Niko Hensel (Hannover Medical School, MHH) for help with the real-time PCR, Johannes N. Greve (MHH) for providing nonmuscle myosin-2, Matthias Ballmaier (Cell sorting facility, MHH) for the FACS experiments and their analysis insights, and Doan Duy Hai Tran (MHH) for helping with the isolation of primary macrophages. We thank Jan Faix, Jonas Scholz, and Thomas Kaufmann (MHH) for providing reagents, plasmids, analysis insights and software. M.P. was supported by the Ministry of Culture and Science of the State North-Rhine Westphalia, Germany (UMMBAS, FKZ 005-2302-0023 and AStaBaK, FKZ 005-2211-0043). G.T. was supported by the German Research Council, Germany (TS169/3-1 and TS169/5-1).

## Author contributions

Conceptualization and project supervision G.T.; compound screening and discovery, H.-J.K., F.K.H., H.O.G., and G.T.; compound synthesis, J.G. and H.-J.K.; cell biological and functional investigations, fluorescence microscopy and analyses, D.K., L.V., A.B, J.D.G., D.S., C.S., P.F., and K.S.; gene expression analyses, D.K., K.S., and P.C.; live cell imaging, spheroid experiments, and analyses, D.K.; crystallographic study and modelling, W.E. and M.P.; protein purification and functional assays, F.K.H., K.E., and P.F.; mouse experiments, B.S., H.B., D.K., I.B., and D.N.; histopathological analysis, K.H. and N.S.; *Drosophila* experiments, M.I.T. and H.R.S.; mass spectrometric analysis, H.B., B.S., and D.N.; NM2A/B microscopy, N.B., D.K., and M.H.T.; data analysis and interpretation, D.K., L.V., A.B, J.D.G, M.Pl., H.S., M.I.T., B.S., A.S., M.A., P.C., and G.T.; organoid experiments, L.V., A.S., and M.A.; TEER assays, M.Pl. and A.N.; supervision, M.H.T., A.N., D.N., H.R.S., M.B., H-J.K., M.P., and G.T.; mechanism conceptualization; D.K. and G.T.; figure preparation; D.K.; writing; G.T. with contributions from D.K., M.A., M.B., H.R.S., H.-J.K., M.P., and A.N.

## Funding

## Competing interests

The authors declare no competing interests.

## Additional information

[1]Institute for Biophysical Chemistry, Hannover Medical School, Hannover, Germany. [2]Institute for Functional Gene Analytics (IFGA), Bonn-Rhein-Sieg University of Applied Sciences, Rheinbach, Germany. [3]Faculty of Chemistry, TU Dresden, Dresden, Germany. [4]Institute of Integrative Cell Biology and Physiology, University of Münster, Münster, Germany. [5]Institute of Cell Biochemistry, Hannover Medical School, Hannover, Germany. [6]Institute of Pharmacology, Hannover Medical School, Hannover, Germany. [7]Department of Cell Physiology and Biophysics, Institute of Cell Biology and Biophysics, Leibniz Universität Hannover, Hannover, Germany. [8]Department of Gastroenterology, Hepatology and Endocrinology, Hannover Medical School, Hannover, Germany. [9]Department of Biology, TU Dresden, Dresden, Germany. [10]Institute for Pathology, Hannover Medical School, Hannover, Germany. [11]Institute of Pathology, KRH Klinikum Nordstadt, Hannover, Germany. [12]Research Core Unit Mass Spectrometry—Metabolomics, Hannover Medical School, Hanover, Germany. [13]Institute for Laboratory Animal Science, Hannover Medical School, Hannover, Germany. [14]SMATHERIA gGmbH—Non-Profit Biomedical Research Institute, Hannover, Germany. ✉e-mail: Tsiavaliaris.Georgios@mh-hannover.de

