## [Transparent Peer Review file · Nature Communications]

An allosteric inhibitor of RhoGAP class-IX myosins suppresses the metastatic features of cancer cells

Corresponding Author: Professor Georgios Tsiavaliaris

Version 1:

Reviewer comments:

Reviewer #1

(Remarks to the Author)

In this manuscript, the authors found a new potential drug "adhibin", a de novo synthesized allosteric inhibitor of RhoGAP class-IX myosins. Adhibin abrogates ATPase and motor function and thus suppresses RhoGTPase-mediated modes of cancer cell metastasis via blocking membrane protrusion formation, disturbing remodelling of cell-matrix adhesions, and disrupting epithelial junction stability et al. Their data provide the basis for developing adhibin as the anti-metastatic cancer therapy drug. This work is interesting. Yet, several questions should be addressed:

1. Although the author has carried out a lot of work in this study, the most crucial thing is to explore the anti-tumor metastasis of adhibin in nude mice to evaluate their efficacy. Therefore, it is necessary to supplement this experiment. In addition, it is recommended that the author use conventional chemotherapy drugs as a control to further evaluate the anti-metastatic effect of adhibin. Finally, the expression changes of key targets related to drug anti-metastasis also need to be detected.
2. When Adhibin significantly represses tumor metastasis in animals, their related side effects, including HE staining of the heart, liver, and kidneys, as well as hematological indicators related to their damage, need to be tested.
3. At the cellular level, Western blot detection is required to detect changes in the expression of Myo9b, ZO-1, and other related genes caused by Adhibin.
4. In this study, Myo9b was the core target of Adhibin. The author should clarify the specific molecular mechanisms by which Myo9b regulates membrane protrusion formation, remodelling of cell matrix adhesions, and epithelial junction stability.

Reviewer #2

(Remarks to the Author)

The class-IX myosins, myosin-9a (Myo9a) and myosin-9b (Myo9b), constitute a unique family of myosin motor proteins that contain a RhoGAP domain in the C-terminal tail to specifically deactivate RhoA. This establishes a connection between class-IX myosins and the RhoA signaling pathway, controlling cell polarization, migration and adhesion. Due to the potential processive movement of the N-terminal motor domain (albeit at a low velocity), this class of myosin motors is capable of self-transporting along actin filaments for the spatial and temporal control of RhoA activity.

In this manuscript, Kyriazi et al. identified a compound called adhibin as an inhibitor of the motor domain of Myo9, decreasing its ATPase activity and processive movement. With the use of different cancer cell lines, they demonstrated that adhibin can suppress cancer cell migration and adhesion, possibly through interference with the RhoA-mediated actin filament dynamics and actomyosin-based contractility. Additionally, adhibin is capable of disassembling cell-cell contacts by disrupting cell-cell junctions. They finally showed that Adhibin only retards morphogenesis of intestinal organoids but has little impact on animal lethality using mouse and fly models.

Although the identification of adhibin as an inhibitor of the motor domain of Myo9 is of interest, three important pieces of data are missing from the manuscript. In the structural characterization of the binding site of adhibin in the motor domain, the authors did not provide solid structural data to demonstrate the binding of adhibin in the motor domain. Instead, they solely

utilized structural docking and modelling to construct the structural model of the motor domain/adhibin complex. Furthermore, the authors did not conduct mutational studies to evaluate the potential adhibin-binding site in the motor domain of Myo9, which is a common strategy to validate the structural model.

In the biochemical characterization of the RhoGAP activity of Myo9, the authors did not perform biochemical experiments to characterize the potential effects of adhibin on the RhoGAP activity of Myo9 towards RhoA. This characterization is crucial for concluding the adhibin-mediated regulation of the RhoA signaling pathway through Myo9. The results regarding the impact of adhibin on the RhoGAP activity of Myo9 should be included in the manuscript; otherwise, the authors cannot exclude the possibility of adhibin-mediated regulation of RhoA via other unclear factors.

In the functional characterization of the potential effects of adhibin in mouse and fly models, the authors demonstrated the impact of adhibin in wild-type animal models. However, it would be beneficial to showcase the positive effects of adhibin in suppressing tumor progression using disease-related animal models. This demonstration would enhance the significance of this compound as an anti-cancer drug. Thus, the lack of the structural, biochemical, and functional data may hinder the publication of this manuscript in Nature Communications.

1. In Figure 1, the authors only showed the adhibin-mediated inhibition of Myo9b, and based on the structural models, some slight differences occur between Myo9a and Myo9b. It would be better to characterize both Myo9a and Myo9b and include the biochemical data about Myo9a in the manuscript since the authors characterized both isoforms at the cellular level.

2. In Figure S1, adhibin seems to impact the protein level of Myo9a but not Myo9b, which suggests that this compound has different effects on Myo9a and Myo9b. Is there a possible explanation for this observation, and would this difference also impact the adhibin-mediated control of the cellular functions of Myo9a and Myo9b?

3. On Page 8, the authors demonstrated the effects of adhibin in controlling cell cycle. What is the potential relationship between cell cycle/proliferation and cell migration for cancer metastasis? The authors may need to summarize all the effects of adhibin and expand the discussion about them in detail.

4. On Page 10, the authors demonstrated the adhibin-mediated regulation of RhoA activity using the pull-down assay. If this regulation is mediated by Myo9, in addition to the RhoA activity assay, the authors may need to further perform the binding assay to check the effects of adhibin on the binding between RhoA and Myo9. This would provide solid evidence regarding the relationship between adhibin and the Myo9-RhoA signaling pathway.

5. On Page 12, the authors demonstrated the inactivation of NM2 with the treatment of adhibin. Since adhibin has no effects on the motor activity of NM2, it would be interesting to compare the structures of the motor domains of different myosins in detail and find out the reason that causes the specific recognition of adhibin by Myo9.

6. Figure S7 summarizes the effects of adhibin on the Myo9-RhoA signaling pathway. However, this figure is too complicated for the readers to follow, and the authors may need to simplify it to only highlight the main conclusions of this manuscript.

Reviewer #3

(Remarks to the Author)

In this manuscript the authors report a new small molecule inhibitor of ATPase activity of myosins IXa and IXb, molecular motors that contain a motor domain in their N-terminus and a Rho-GAP domain in their C-terminal portions. This inhibitor shows high selectivity for inhibiting ATPase and motor activity of class IX myosins compared with the other myosin classes tested and exhibits relatively low toxicity in cells and in vivo. The authors then demonstrate that the selected inhibitor compound (adhibin) interferes with cell migration and adhesion and cell cycle progression in several cancer cell lines. They find that the drug treatment of the cancer cell lines leads to changes in the overall cell shape and actin organization, a reduction in actin stress fibers and a decrease in the size of focal adhesions. These latter observations (fewer stress fibers and focal adhesions) are consistent with a possible decrease in Rho activity. Indeed, the authors show a slight decrease in the amount of active Rho in inhibitor-treated cells.

The authors propose that inhibition of the myosin IX motor domain activity may somehow result in the enhancement of its Rho-GAP activity or the abnormal spatial redistribution of its Rho-GAP activity, leading to inactivation of Rho in cells. This is indeed one of the possible explanations of their observations, and a very interesting finding from the standpoint of understanding the roles of myosin IX in cells. However, the manuscript does not provide a definitive proof that it is indeed the inhibition of myosin IX, and not an effect on some unrelated target in cells, that leads to the observed morphological and functional changes in cells. If the authors had performed a knockdown of myosins IXa and IXb and then demonstrated that the cells lacking these myosins do not respond to the inhibitor treatment with the same changes in actin and focal adhesion organization, that would provide the most convincing support for their hypothesis.

As is, the manuscript is well-written and contains a large amount of data; I believe it provides very interesting and valuable information to the cell biology community. If the knockdown experiments discussed above are too challenging, it may be worthwhile to simply discuss the potential alternative explanations in the paper.

I think it is also important to point out some of the findings that contradict the authors' hypothesis – for example, on pg. 8 they mention previous findings on “Myo9b-depleted cells, which similarly failed to spread and establish a polarized shape, as well as to collectively migrate”. If the previously published findings demonstrate that Myo9b depletion resulted in the same effects on cell migration as Myo9b inhibition, this would argue against the aberrant Myo9 RhoGAP activity being responsible for the inhibitor effects on cell migration and cell shape. Similarly, the authors demonstrate that inhibitor treatment disrupts cell-cell contacts and compare these findings with the previously published observations of junctional disruption in Myo9a/9b-depleted cells. The authors discuss the earlier findings (pg.18) and state that they are consistent with their proposed mechanism of action of adhibin (dysregulated Rho activity) but that seems like a very broad/non-specific explanation for how a complete loss of the Myo9 Rho-GAP (via knockout/knockdown in previous studies) can have the same effects as the dysregulated Myo9 Rho-GAP activity (this paper). It may be good to discuss how these findings can be reconciled with each other.

Additional comments:

References are somehow split into two sections (1-109 before the Materials and Methods, and the rest after).

The following section appears to refer to missing or mislabeled figures: “Progeny from adhibin-treated parents showed normal nervous system development during embryogenesis (Figure 7I) and photoreceptor axonal projection were properly established in the brain of adhibin-treated larvae (Figure 7J). Additionally, in adult brains, there were no apparent changes observed in the structure of the learning and memory centre or the tight junctions of the blood-brain barrier (Figure 7K,L).” It looks like this may be referring to Extended Fig. 6 rather than Fig. 7.

What is meant by the organoid growth in Extended Fig. 6E? This is explained in the main text (pg.15) but not in the figure legend or on the axis label.

Methods:

Cell lines – what is ATTG? Should probably be ATCC.

How was F/G actin ratio determined? Some type of fractionation/extraction procedure?

For the myosin purification procedures (multiple myosins – “All other myosins were purified from native tissue or recombinant from the following organismal sources: *D. discoideum* (Dd), *O. coniculus* (Oc), *S. scrofa* (Ss), *R. norvegicus* (Rn), *C. elegans* (Ce), *H. sapiens* (Hs). Motor domain constructs of, myosin-2, myosin-5a,-5b, Ce myosin-9/Rn myosin-9b, were prepared as described previously 113–119”), several references are included but a brief description of the methods is not provided. Perhaps adding a table listing specific myosins, the figures where they were used, the source organism, and the tag or absence of the tag along with the corresponding reference would be helpful. Otherwise it is unclear how the recombinant myosin expression constructs were cloned, whether any affinity tags were used, whether most myosins listed were expressed in *Dictyostelium* or in other organisms etc. For example, ref. 119 describes purification of myosin IX using the baculovirus system, not an organismal source. Given the central importance of the myosin inhibition experiments to the main topic of this paper, it is important that this information is described well.

Rho activity measurements – Extended data Figure 3, panels Q and R. It is common to compare the amount of active Rho (bound to Rhotekin beads) with the amount of total Rho in the input lysate. Total Rho is not shown in panel Q, and it is unclear whether the Rho amounts used in panel R were normalized in any way (equal amounts of cell lysates used as inputs? Equal amounts of total Rho? Etc.).

Pg.7 – “adhibin displays the lowest cytotoxic” – this sentence is incomplete.

Version 2:

Reviewer comments:

Reviewer #1

(Remarks to the Author)

The focus of drug development is on its functionality and efficacy, especially evaluating drug efficacy at the animal level is crucial. The author raised some difficulties indicating that it is difficult to perform the evaluation of the anti-tumor effect of the drug at the animal level, and therefore the development value of the drug cannot be confirmed. I therefore suggest that the author must still complete the first question I raised. In addition, the author also needs to use a common chemotherapy drug as a control for comprehensive evaluation.

Reviewer #2

(Remarks to the Author)

The manuscript has been strengthened with the additional structural modelling and functional data, and the reviewer supports the publication of this work in Nature Communications.

Reviewer #3

(Remarks to the Author)

The authors have addressed most of my concerns.

I do have a question/confusion regarding one of the experiments they have added in response to my questions.

Specifically, the authors have performed an siRNA-mediated knockdown of Myo9B in B16-F1 and A549 cells to compare cells lacking Myo9B with the adhibin-treated cells (supp. Figure 5 and 7). The siRNA described in the Materials and Methods section are pre-designed siRNAs from Qiagen that target human Myo9b. B16-F1 cells are mouse cells. Which siRNAs were used to target Myo9b in this cell line? In addition, Materials and methods provide information for siRNAs numbered 3, 4, and 5 while Supp. Figure 5 shows the results for siRNAs numbered 2, 3, and 5. I recommend that the authors carefully cross-check the siRNA information provided in the Materials and methods and the data.

Minor comments:

Supp. Figure 3 – the title reads “Adhibin impairs surface adherens”, should probably be “surface adhesion”

Supp. Figure 4Q – quantification of the active Rho presents a decrease in 3 cell lines treated with 25 uM adhibin. However, the corresponding Western blot shown later in the supplemental data file shows treatment with 10um drug for the B16 cells.

Version 3:

Reviewer comments:

Reviewer #1

(Remarks to the Author)

I don't care about the data (Figure 1) provided by the author, I just care about the efficacy of the drug and its potential application prospects. So far, the author has refused to conduct critical research in this area at the animal level, thus unable to effectively evaluate the anti-tumor ability of drugs. I therefore do not agree to publish the paper under the current circumstances unless it can effectively evaluate the drug efficacy at the animal level.

Reviewer #2

(Remarks to the Author)

No further comments.

Reviewer #3

(Remarks to the Author)

All of my comments have been addressed. I recommend the manuscript for publication.

made.

Point-by-point answers to the reviewers

We thank the reviewers and the editorial board for the friendly evaluation of our manuscript, the constructive comments and the valuable suggestions to improve the quality of our work. We have addressed all points raised by the referees as requested and included new and additional data, controls, validation experiments with quantifications where appropriate. All major modifications made to the manuscript are highlighted. We also provide a clean copy of the manuscript to the submission.

Reviewer #1 (Remarks to the Author):

In this manuscript, the authors found a new potential drug “adhibin”, a de novo synthesized allosteric inhibitor of RhoGAP class-IX myosins. Adhibin abrogates ATPase and motor function and thus suppresses RhoGTPase-mediated modes of cancer cell metastasis via blocking membrane protrusion formation, disturbing remodelling of cell-matrix adhesions, and disrupting epithelial junction stability et al. Their data provide the basis for developing adhibin as the anti-metastatic cancer therapy drug. This work is interesting. Yet, several questions should be addressed:

1. Although the author has carried out a lot of work in this study, the most crucial thing is to explore the anti-tumour metastasis of adhibin in nude mice to evaluate their efficacy. Therefore, it is necessary to supplement this experiment. In addition, it is recommended that the author use conventional chemotherapy drugs as a control to further evaluate the anti-metastatic effect of adhibin. Finally, the expression changes of key targets related to drug anti-metastasis also need to be detected.

This comment of the reviewer is undoubtedly an important experimental approach for testing *in vivo* the anti-tumour efficacy of the compound. However, the permission for animal experiments obtained so far by the Specialised Department of Animal Welfare Service of Lower Saxony allowed only for toxicity and pharmacokinetic investigations with a defined number of animals, which are part of the manuscript (see Material and Methods). We were also able to obtain preliminary pharmacokinetic data including drug uptake and distribution in serum and organs (**Revision Figure 1**).

Revision Figure 1. Normalized peak area of heart, liver, spleen, kidney, lung (counts/ organ wet weight), plasma (counts) and serum (concentration-peak area) in female and male mice, over time showing drug uptake, distribution, and degradation, including adhibin concentration in the plasma and serum at defined time points after intraperitoneal injection.

The data in **Revision Figure 1** is intended for use only in the revision process, since the experiments are still ongoing and for a quantitative analysis, a second round of animal experiments is necessary. Pharmacokinetic data are intended to be published elsewhere. **Therefore we request the editorial board to redact these preliminary data from the peer review file.**

With regard to the reviewer's recommendation of evaluating anti-metastatic effects of adhibin in nude mice, we aim to perform drug efficacy experiments with humanized rat models with our collaborators as soon as the Animal Welfare Service of Lower Saxony has given us the permission. Due to highly restrictive laws, applications for drug testing experiments in animals are not easily granted in Germany, take long, and can even be rejected. Yet, we cannot foresee, when we will be able to continue with these experiments. Just for reference, the animal experiments described herein took 36 months till we the permission was granted. As recommended by the reviewer, we have already started studying potential adhibin-induced expression changes of key targets using high-throughput screening approaches and next generation sequencing. Due to ongoing work, extensiveness and complexity of such gene analyses, we intend to publish the new findings in a separate work.

However, we tested the compound in a more physiologically relevant environment. To mimic tissue and micro-tumour conditions, we created spheroids of approx. 200 µm size from mouse fibroblasts, which are widely used as artificial tumour models in cancer research, often employed as a suitable system in anticancer drug development as demonstrated¹. We tested the efficacy of adhibin to potentially interfere with tumour growth and metastasis by studying spheroid formation and growth on the basis of critical parameters such as 2D/3D cell migration, spheroid assembly and disassembly. These data have now been included into the manuscript and are part of a new main figure (**Figure 9A-F**).

2. When adhibin significantly represses tumour metastasis in animals, their related side effects, including HE staining of the heart, liver, and kidneys, as well as hematological indicators related to their damage, need to be tested.

We had performed this analysis before but did not include the data in the manuscript, since the histopathological evaluation of the organs and bone marrow revealed no damage by adhibin. Please find below the organ staining of control and adhibin-treated animals as a separate figure (**Revision Figure 2**). **These data have not been included in the manuscript and should be redacted from the peer review file when published.**

Revision Figure 2. HE stainings of cerebrum, cerebellum, heart, lung, liver, spleen, pancreas, gut, kidney and bone marrow in control and drug treated mice at highest tolerable doses as imposed by the Specialised Department of Animal Welfare Service of Lower Saxony.

Additionally, we demonstrate the successful uptake of the drug in the *Drosophila* model, both in adult flies and larvae by mass spectrometry. The table below (**Supplementary Table 4**) has been included as supplementary table in the manuscript. Methods and analysis have been added as additional chapter “Liquid chromatography coupled mass spectrometry”.

3. At the cellular level, Western blot detection is required to detect changes in the expression of Myo9b, ZO-1, and other related genes caused by Adhibin.

The reviewer might have missed this information: the levels of Myo9a, Myo9b, and phosphorylated NM2s were shown in the initial version of the manuscript, and are also part of our revised manuscript: **Suppl. Figure 2 (panels E-H), Suppl. Figure 4 (panels A,B), Figure 7 (panels E,F), and Figure 6 (panels E-F).**

We followed the advice of the reviewer and analyzed the levels of other related proteins in the presence of adhibin by western blot. This includes:

1. the NM2-Light Chain phosphatase (p-MLCP), which is phosphorylated downstream of Rho by ROCK. We show this data as additional panel in **Figure 7 (panel G)**:
2. paxillin, vinculin, VASP and ZO-1, which are involved in cell-cell and cell-substrate adhesion. We show this data in (**Supplementary Figure 8**).

The results section has been modified accordingly and the discussion has been extended as highlighted in the revised manuscript.

4. In this study, Myo9b was the core target of adhibin. The author should clarify the specific molecular mechanisms by which Myo9b regulates membrane protrusion formation, remodelling of cell matrix adhesions, and epistemic junction stability.

Membrane protrusion formation requires the tight regulation of signaling networks regulating actin dynamics, amongst these the cascades that a) activate Rac and Cdc42, which drive actin filament polymerization, b) activate Rho/ROCK, which stimulates through LIMK1/2 activation and cofilin inactivation the assembly and stability of filamentous actin structures and c) locally inhibit Rho activity, which is realized through the recruitment of the RhoGAP Myo9b. Thus, Myo9b is an essential modulator of the cross talk between Rac and Rho important to control protrusion dynamics and support cell migration. Its inhibition by adhibin perturbs protrusion dynamics and suppresses the migratory properties of the cells.

These aspects have now been included as part of the results (page 13) and discussion in the revised manuscript highlighted in red (pages 21-24).

Reviewer #2 (Remarks to the Author):

Comments to the authors

The class-IX myosins, myosin-9a (Myo9a) and myosin-9b (Myo9b) constitute a unique family of myosin motor proteins that contain a RhoGAP domain in the C-terminal tail to specifically deactivate RhoA. This establishes a connection between class-IX myosins and the RhoA signaling pathway, controlling cell polarization, migration and adhesion. Due to the potential processive movement of the N-terminal motor domain (albeit at a low velocity), this class of myosin motors is capable of self-transporting along actin filaments for the spatial and temporal control of RhoA activity. In this manuscript, Kyriazi et al. identified a compound called adhibin as an inhibitor of the motor domain of Myo9, decreasing its ATPase activity and processive movement. With the use of different cancer cell lines, they demonstrated that adhibin can suppress cancer cell migration and adhesion, possibly through interference with the RhoA-mediated actin filament dynamics and actomyosin-based contractility. Additionally, adhibin is capable of disassembling cell-cell contacts by disrupting cell-cell junctions. They finally showed that Adhibin only retards morphogenesis of intestinal organoids but has little impact on animal lethality using mouse and fly models.

1. Although the identification of adhibin as an inhibitor of the motor domain of Myo9 is of interest, three important pieces of data are missing from the manuscript. In the structural characterization of the binding site of adhibin in the motor domain, the authors did not provide solid structural data to demonstrate the binding of adhibin in the motor domain. Instead, they solely utilized structural docking and modelling to construct the structural model of the motor domain/adhibin complex. Furthermore, the authors did not conduct mutational studies to evaluate the potential adhibin-binding site in the motor domain of Myo9, which is a common strategy to validate the structural model.

This is a justified comment by the reviewer. However, it must be noted that the crystallization of myosin motor domains is – despite years of research – a major obstacle in the field as exemplified by the few X-ray structures available in the protein data base covering a moderate selection of myosin isoforms from only four classes out of twelve found in humans or mammals. In our previous and current work, the crystallization of the class-II model myosin from *Dictyostelium* in combination with molecular modelling and molecular dynamics simulation evolved as practicable and quite successful approach to identify potential binding sites of various inhibitors and modulators²⁻⁵ and uncover structural insights into the molecular mechanism of force production^{6,7} and its inhibition/modulation by small chemical compounds⁸⁻¹²). Although the binding pocket of adhibin in Myo9 may differ from the modelling prediction, it must be noted that a) the NADH-coupled assay, which quantifies ATP hydrolysis/turnover of the myosins and b) the *in vitro* motility assay with recombinant myosin motor domain constructs as used in this study, are both standard functional assays unambiguously demonstrating that adhibin inhibits the actin-activated ATPase reaction and the motor activity of the Myo9s through binding to the motor domain.

With respect to the introduction of mutations to validate experimentally the binding site, we experienced difficulties in obtaining stable protein for biochemical analysis. The introduction of mutations in such a critical region of the motor domain is not expected to facilitate purification, stability, and analysis of the protein. We therefore have chosen an *in silico* approach based on homology modelling and molecular docking to predict the preferred binding site of adhibin in Myo9a and Myo9b, which would provide the structural basis for its increased inhibitory potency towards class-IX myosins. We clarified this aspect by adding new data to the manuscript (**Supplementary Figure 1**). The additional information has been highlighted in the manuscript (page 6-8, 20-21).

Because of the high sequence similarity between class-V and class-IX myosin motor domains ($\approx 77\%$), we took the pre-power stroke state crystal structure of Ggmyosin-5a as template for the generation of the

Myo9 homology models. Additionally, we generated a homology model in the post-rigor state to account for conformational differences between the states affecting the geometry of potential binding sites. Initial blind docking of adhibin to the Myo9 homology models including the crystal structures of other myosins *DdMyo1E*, *DdMyo2* and *GgMyo5a*, did not produce clear hits of energetically favorable protein-ligand interactions justifying the presence of potential binding sites. To improve our docking results, we performed molecular dynamics (MD) simulations and obtained an ensemble of conformational states reflecting the dynamics of the pre-power-stroke state and post-rigor state of the myosin structure. Using ensemble-based docking^{13,14} and position clustering that differentiates between stable and unstable interactions on the basis of root mean square deviation (RMSD) and the corresponding Gibbs-energies, we identified suitable ligand geometries assigning the inner cleft region in the myosin motor domain as the preferable binding site for adhibin with Gibbs energies of $\Delta G \sim -8$ kcal/mol. We tested the validity of approach by performing the same docking algorithm with *DdMyo2* and PBP as ligand, for which the binding pocket has been previously described (Fedorov et al. 2009). This approach yielded the identification of the same reported binding pocket with a ΔG value of the protein-ligand interaction of ~ -8 kcal/mol. This is consistent with the value obtained from the crystal structure. The subsequent ensemble-based docking with the *DdMyo1E* structure yielded a slightly different position of the ligand compared to Myo9b. The binding pocket of adhibin in *DdMyo1E* was found to be located between the blebbistatin and PBP binding site. The calculated Gibbs-energy is -7.2 kcal/mol. In the case *DdMyo2* and *DdMyo5a*, the adhibin binding pocket is again shifted compared to the position found in *DdMyo1E* but closer to the PBP binding site. The corresponding Gibbs-energy is $\Delta G = -6.2$ kcal/mol indicating a very unstable interaction. In summary, the calculated ΔG -values reflect in good agreement the experimentally determined binding affinities of adhibin to the respective myosins. To evaluate the stability of adhibin in the binding pocket, we performed accelerated MD-simulation (aMD), a method that enables an enhanced conformational space sampling between distinct states of the potential surface by reducing energy barriers between them. Only those energy basins are raised that are below a certain threshold level, while those above this level remain unaffected. As a result, sampling is highly accelerated and conformational states become accessible that cannot be simulated using standard MD simulations. The 40 ns aMD-simulations with the myosin motor domain structures in complex with adhibin reveal for *RnMyo9b* constantly small RMSD values ~ 2 Å on average over time for the ligand position. For the other myosins, adhibin coordination within the binding pocket is less stable. The RMSD-values distribute widely over time populating on average distances between 5 and 10 Å for *DdMyo1E*, 15 to 80 Å for *DdMyo2*, and 15 to 40 Å for *GgMyo5a*. **This computational work is part of a broader structural and computational study that is still under investigation. We hope the reviewer and the editorial board understand our intention to thoroughly address this aspect in more detail and include the data in a separate structural study for publication in a specialized journal. Therefore we request the editorial board to redact this information from the peer review file.**

2. In the biochemical characterization of the RhoGAP activity of Myo9, the authors did not perform biochemical experiments to characterize the potential effects of adhibin on the RhoGAP activity of Myo9 towards RhoA. This characterization is crucial for concluding the adhibin-mediated regulation of the RhoA signaling pathway through Myo9. The results regarding the impact of adhibin on the RhoGAP activity of Myo9 should be included in the manuscript; otherwise, the authors cannot exclude the possibility of adhibin-mediated regulation of RhoA via other unclear factors.

We fully agree with the reviewer that characterizing potential effects of adhibin on Myo9 RhoGAP activity towards RhoA through a direct assay with purified proteins is an additional approach for gaining additional insights into the mechanism of action. However, biochemical assays would require the full-length protein, which we have not been able to obtain until now. Even the motor domain constructs used in this study

are instable over long-term storage, do not resist freeze-thawing, and lose most of the ATPase activity within 36 hours after purification. Therefore, all biochemical data presented here were obtained from several preparations with freshly purified protein immediately used for the experiments within 6-24 hours after purification. Although the purification of full-length Myo9a has been reported, the amounts obtained are in the low microgram range, which restricts the experimental framework to e.g. single molecule and electron microscopic studies¹⁵. Unfortunately, we are not able to prepare full-length protein for studying adhibin on the RhoGAP activity of Myo9 directly.

However, we were able to test the drug on recombinant RhoA from mouse. Shortly, we generated expression vectors encoding GST-tagged RhoA. The RhoA gene was obtained by amplification from cDNA isolated from B16-F1 cells. We followed the purification procedure described previously¹⁶ and tested the basal GTPase activity of RhoA in the presence of adhibin (**12**) and its analogue (compound **5**) as shown in **Revision Figure 3**. There is no apparent effect of the drug on RhoA GTPase activity. **We request the editorial board to redact this information from the peer review file, since we are investigating this aspect further.**

Revision Figure 3. Effect of adhibin on RhoA. RhoA activity (s^{-1}) in the presence of increasing concentrations of adhibin or analogue 5. RhoA was purified in Rosetta 2 with a GST tag. The P_i -release was quantified using the MDCC sensor described in Franz et al, 2020¹⁷, and measured using a plate reader at 405 nm excitation and 472nm absorption.

3. In the functional characterization of the potential effects of adhibin in mouse and fly models, the authors demonstrated the impact of adhibin in wild-type animal models. However, it would be beneficial to showcase the positive effects of adhibin in suppressing tumour progression using disease-related animal models. This demonstration would enhance the significance of this compound as an anti-cancer drug. Thus, the lack of the structural, biochemical, and functional data may hinder the publication of this manuscript in Nature Communications.

We appreciate the suggestion of the reviewer to conduct tumour animal models. We have addressed this point already with the answers to question 1 of reviewer #1, referring to additional data, which are included in **Revision Figure 1** and as new panels part of Figure 9 (**Figure 9A-F**). Descriptions and interpretations of data are found in the main text of the revised manuscript (page 18 and discussion).

4. In Figure 1, the authors only showed the adhibin-mediated inhibition of Myo9b, and based on the structural models, some slight differences occur between Myo9a and Myo9b. It would be better to characterize both Myo9a and Myo9b and include the biochemical data about Myo9a in the manuscript since the authors characterized both isoforms at the cellular level.

This is an important point. Due to the lack of Myo9a protein, we addressed this aspect by testing the inhibitory potency of adhibin towards the only class-IX myosin present in *C. elegans*, namely CeMyo9, which displays high sequence similarity to Myo9a as sequence alignments show¹⁸. However, we provided additional information on how adhibin induced defects relate to Myo9a and Myo9b (Discussion: pages 21-24).

5. In Figure S1, adhibin seems to impact the protein level of Myo9a but not Myo9b, which suggests that this compound has different effects on Myo9a and Myo9b. Is there a possible explanation for this observation, and would this difference also impact the adhibin-mediated control of the cellular functions of Myo9a and Myo9b?

The altered Myo9a levels may indeed be related to adhibin. Currently, we do not have an explanation for this. It might be related to difference in the expression levels of the two isoforms between the cell lines, which might be differentially affected by adhibin. The different localization highlights the distinct functions of the isoforms, which could be regulated through distinct pathways and feedback mechanisms. We have first indications of a potential feedback control of the myosins that is currently being investigated.

6. On Page 8, the authors demonstrated the effects of adhibin in controlling cell cycle. What is the potential relationship between cell cycle/proliferation and cell migration for cancer metastasis? The authors may need to summarize all the effects of adhibin and expand the discussion about them in detail.

We thank the reviewer for this comment. We had already provided an explanation in chapter “**Adhibin interferes with Rho-mediated actin dynamics, actomyosin-based contractility, and cytokinesis**” and added additional information as highlighted in the results and discussion section (pages 12, 13, 15, 22, 23)

7. On Page 10, the authors demonstrated the adhibin-mediated regulation of RhoA activity using the pull-down assay. If this regulation is mediated by Myo9, in addition to the RhoA activity assay, the authors may need to further perform the binding assay to check the effects of adhibin on the binding between RhoA and Myo9. This would provide solid evidence regarding the relationship between adhibin and the Myo9-RhoA signaling pathway.

The reviewer makes an important comment. Binding of Myo9s to RhoA through the GAP domain has been demonstrated previously together with the X-ray structure of the Myo9b RhoGAP domain and by structural modelling of the Myo9b RhoGAP domain in complex with RhoA¹⁹. To study the effect of adhibin on the RhoGAP activity of Myo9 towards RhoA through a direct binding approach, we would require the full-length variants of the proteins, which we unfortunately have not been able to obtain until now.

8. On Page 12, the authors demonstrated the inactivation of NM2 with the treatment of adhibin. Since adhibin has no effects on the motor activity of NM2, it would be interesting to compare the structures of the motor domains of different myosins in detail and find out the reason that causes the specific recognition of adhibin by Myo9.

We thank the reviewer for this comment. The answer to this concern has been provided above with the answer to question 1 from the same reviewer.

9. Figure S7 summarizes the effects of adhibin on the Myo9-RhoA signaling pathway. However, this figure is too complicated for the readers to follow, and the authors may need to simplify it to only highlight the main conclusions of this manuscript.

We agree with the reviewer that the scheme contains a lot of information. We have modified it and interpreted the most relevant aspects in the discussion (pages 21 ff.). This figure is now a main figure (Figure 10).

Reviewer #3 (Remarks to the Author):

In this manuscript, the authors report a new small molecule inhibitor of ATPase activity of myosins IXa and IXb, molecular motors that contain a motor domain in their N-terminus and a Rho-GAP domain in their C-terminal portions. This inhibitor shows high selectivity for inhibiting ATPase and motor activity of class IX myosins compared with the other myosin classes tested and exhibits relatively low toxicity in cells and in vivo. The authors then demonstrate that the selected inhibitor compound (adhibin) interferes with cell migration and adhesion and cell cycle progression in several cancer cell lines. They find that the drug treatment of the cancer cell lines leads to changes in the overall cell shape and actin organization, a reduction in actin stress fibers and a decrease in the size of focal adhesions. These latter observations (fewer stress fibers and focal adhesions) are consistent with a possible decrease in Rho activity. Indeed, the authors show a slight decrease in the amount of active Rho in inhibitor-treated cells.

1. The authors propose that inhibition of the myosin IX motor domain activity may somehow result in the enhancement of its Rho-GAP activity or the abnormal spatial redistribution of its Rho-GAP activity, leading to inactivation of Rho in cells. This is indeed one of the possible explanations of their observations, and a very interesting finding from the standpoint of understanding the roles of myosin IX in cells. However, the manuscript does not provide a definitive proof that it is indeed the inhibition of myosin IX, and not an effect on some unrelated target in cells, that leads to the observed morphological and functional changes in cells. If the authors had performed a knockdown of myosins IXa and IXb and then demonstrated that the cells lacking these myosins do not respond to the inhibitor treatment with the same changes in actin and focal adhesion organization, that would provide the most convincing support for their hypothesis. As is, the manuscript is well-written and contains a large amount of data; I believe it provides very interesting and valuable information to the cell biology community. If the knockdown experiments discussed above are too challenging, it may be worthwhile to simply discuss the potential alternative explanations in the paper.

We thank the reviewer for the positive feedback and the valuable comments/suggestions to improve the quality of our manuscript. We have focused our investigations on providing further experimental support of a Myo9 dependent mechanism through which adhibin acts. Indeed, silencing experiments can be quite tricky and often as difficult to interpret as with drug interferences or knockouts. Nevertheless, we followed the advice of the reviewer and successfully silenced *MYO9B* in B16-F1 and A549 cells using a mixture of siRNAs. The new findings have been added to the revised manuscript in the results (**Supplementary Figure 5; Supplementary Figure 7E-H**) and discussion section as highlighted (pages 10, 13f).

2. I think it is also important to point out some of the findings that contradict the authors' hypothesis – for example, on pg. 8 they mention previous findings on “Myo9b-depleted cells, which similarly failed to spread and establish a polarized shape, as well as to collectively migrate”. If the previously published findings demonstrate that Myo9b depletion resulted in the same effects on cell migration as Myo9b inhibition, this would argue against the aberrant Myo9 RhoGAP activity being responsible for the inhibitor effects on cell migration and cell shape. Similarly, the authors demonstrate that inhibitor treatment disrupts cell-cell contacts and compare these findings with the previously published observations of junctional disruption in Myo9a/9b-depleted cells. The authors discuss the earlier findings (pg.18) and state that they are consistent with their proposed mechanism of action of adhibin (dysregulated Rho activity) but that seems like a very broad/non-specific explanation for how a complete loss of the Myo9 Rho-GAP (via knockout/knockdown in previous studies) can have the same effects as the dysregulated Myo9 Rho-GAP activity (this paper). It may be good to discuss how these findings can be reconciled with each other.

This is a critical point raised by the reviewer and we apologize, if we were not clear enough with our argumentation. We revised this aspect as part of an extended results (pages 12, 13, 15) and new discussion part in the manuscript as highlighted in red.

Additional comments:

References are somehow split into two sections (1-109 before the Materials and Methods, and the rest after).

We have corrected this.

The following section appears to refer to missing or mislabelled figures: “Progeny from adhibin-treated parents showed normal nervous system development during embryogenesis (Figure 7I) and photoreceptor axonal projection were properly established in the brain of adhibin-treated larvae (Figure 7J). Additionally, in adult brains, there were no apparent changes observed in the structure of the learning and memory centre or the tight junctions of the blood-brain barrier (Figure 7K,L).” It looks like this may be referring to Extended Fig. 6 rather than Fig. 7.

We thank the reviewer. It has been corrected.

What is meant by the organoid growth in Extended Fig. 6E? This is explained in the main text (pg.15) but not in the figure legend or on the axis label.

The information has been added.

Cell lines – what is ATTG? Should probably be ATCC.

It has been corrected.

How was F/G actin ratio determined? Some type of fractionation/extraction procedure

We apologise for having missed to describe of how the G/F-actin ratio was quantified. Details have now been included in the material and methods section.

For the myosin purification procedures (multiple myosins – “All other myosins were purified from native tissue or recombinant from the following organismal sources: *D. discoideum* (Dd), *O. coniculus* (Oc), *S. scrofa* (Ss), *R. norvegicus* (Rn), *C. elegans* (Ce), *H. sapiens* (Hs). Motor domain constructs of myosin-2, myosin-5a,-5b, Cemyosin-9/Rnmyosin-9b, were prepared as described previously 113–119”, several references are included but a brief description of the methods is not provided. Perhaps adding a table listing specific myosins, the figures where they were used, the source organism, and the tag or absence of the tag along with the corresponding reference would be helpful. Otherwise it is unclear how the recombinant myosin expression constructs were cloned, whether any affinity tags were used, whether most myosins listed were expressed in *Dictyostelium* or in other organisms etc. For example, ref. 119 describes purification of myosin IX using the baculovirus system, not an organismal source. Given the central importance of the myosin inhibition experiments to the main topic of this paper, it is important that this information is described well.

We thank the reviewer for this useful comment. A table has been included in the Materials and Methods section (page 27).

Rho activity measurements – Extended data Figure 3, panels Q and R. It is common to compare the amount of active Rho (bound to Rhotekin beads) with the amount of total Rho in the input lysate. Total Rho is not shown in panel Q, and it is unclear whether the Rho amounts used in panel R were normalized in any way (equal amounts of cell lysates used as inputs? Equal amounts of total Rho? Etc.).

We thank the reviewer for this comment. The total amounts of RhoA have now been included and the experiment extended to MLE-12 cells. The blots have now been transferred to the supplementary information file in the section “western blots” and replaced by **Supplementary Figure 4Q** showing the changes in the levels of total RhoA and RhoA•GTP after overnight drug exposure.

Pg.7 – “adhibin displays the lowest cytotoxic” – this sentence is incomplete.

We have corrected this.

REFERENCES (point-by-point answers, not corresponding to numbering in the main manuscript)

1. Yamada, K. M. & Cukierman, E. Modeling Tissue Morphogenesis and Cancer in 3D. *Cell* **130**, 601–610 (2007).
2. Fedorov, R. *et al.* The mechanism of pentabromopseudilin inhibition of myosin motor activity. *Nat Struct Mol Biol* **16**, 80–88 (2009).
3. Chinthalapudi, K. *et al.* Mechanism and specificity of pentachloropseudilin-mediated inhibition of myosin motor activity. *Journal of Biological Chemistry* **286**, 29700–29708 (2011).
4. Ewert, W., Franz, P., Tsiavaliaris, G. & Preller, M. Structural and Computational Insights into a Blebbistatin-Bound Myosin•ADP Complex with Characteristics of an ADP-Release Conformation along the Two-Step Myosin Power Stroke. *Int J Mol Sci* **21**, 7417 (2020).
5. Radke, M. B. *et al.* Small molecule-mediated refolding and activation of myosin motor function. *Elife* **3**, (2014).
6. Diensthuber, R. P., Hartmann, F. K., Kathmann, D., Franz, P. & Tsiavaliaris, G. Switch-2 determines Mg²⁺+ADP-release kinetics and fine-tunes the duty ratio of Dictyostelium class-1 myosins. *Front Physiol* **15**, (2024).
7. Preller, M. & Holmes, K. C. The myosin start-of-power stroke state and how actin binding drives the power stroke. *Cytoskeleton* **70**, 651–660 (2013).
8. Franz, P., Ewert, W., Preller, M. & Tsiavaliaris, G. Unraveling a Force-Generating Allosteric Pathway of Actomyosin Communication Associated with ADP and Pi Release. *Int J Mol Sci* **22**, 104 (2020).
9. Diensthuber, R. P. *et al.* Kinetic mechanism of *Nicotiana tabacum* myosin-11 defines a new type of a processive motor. *The FASEB Journal* **29**, 81–94 (2015).
10. Manstein, D. J. & Preller, M. Small Molecule Effectors of Myosin Function. in 61–84 (2020). doi:10.1007/978-3-030-38062-5_5.
11. Chinthalapudi, K., Heissler, S. M., Preller, M., Sellers, J. R. & Manstein, D. J. Mechanistic insights into the active site and allosteric communication pathways in human nonmuscle myosin-2C. *Elife* **6**, (2017).
12. Preller, M., Chinthalapudi, K., Martin, R., Knölker, H.-J. & Manstein, D. J. Inhibition of Myosin ATPase Activity by Halogenated Pseudilins: A Structure–Activity Study. *J Med Chem* **54**, 3675–3685 (2011).
13. Velazquez, H. A. *et al.* Ensemble docking to difficult targets in early-stage drug discovery: Methodology and application to fibroblast growth factor 23. *Chem Biol Drug Des* **91**, 491–504 (2018).
14. Singh, V. K. & Coumar, M. S. Ensemble-based virtual screening: identification of a potential allosteric inhibitor of Bcr-Abl. *J Mol Model* **23**, (2017).

15. Saczko-Brack, D. *et al.* Self-organization of actin networks by a monomeric myosin. *Proceedings of the National Academy of Sciences* **113**, (2016).
16. Lin, Y., Watanabe-Chailland, M. & Zheng, Y. Protocol for structural and biochemical analyses of RhoA GTPase. *STAR Protoc* **2**, 100541 (2021).
17. Franz, P. *et al.* A thermophoresis-based biosensor for real-time detection of inorganic phosphate during enzymatic reactions. *Biosens Bioelectron* **169**, 112616 (2020).
18. Wallace, A. G., Raduwan, H., Carlet, J. & Soto, M. C. The RhoGAP HUM-7/myo9 integrates signals to modulate RHO-1/RhoA during embryonic morphogenesis in *caenorhabditis elegans*. *Development (Cambridge)* **145**, (2018).
19. Kong, R. *et al.* Myo9b is a key player in SLIT/ROBO-mediated lung tumor suppression. *Journal of Clinical Investigation* **125**, 4407–4420 (2015).

Point-by-point answers to the reviewers

We thank the editorial board and the reviewers for the kind reevaluation of our manuscript and the opportunity to address the remaining issues in a second round of revision. We provide a revised manuscript with supplementary experiments and a point-by-point response to the editors' and the reviewers' comments.

■ Answers to the reviewers:

Reviewer 1: The focus of drug development is on its functionality and efficacy, especially evaluating drug efficacy at the animal level is crucial. The author raised some difficulties indicating that it is difficult to perform the evaluation of the anti-tumor effect of the drug at the animal level, and therefore the development value of the drug cannot be confirmed. I therefore suggest that the author must still complete the first question I raised. In addition, the author also needs to use a common chemotherapy drug as a control for comprehensive evaluation.

- We appreciate the concern of the reviewer and we followed the suggestion to provide data on expression changes of key targets related to metastasis. This includes a selection of cytoskeleton-related targets RHOA, MYO9A, MYO9B, PXN, VCL, VASP, ACT β , MYH9, as well as proto-oncogenic and tumor-suppressive targets of metastasis, such as BRAF, CDK4, EGFR, KEAP1, KRAS, PIK3CA, PTEN, PTPRD, TP53¹⁻⁵ (**2nd Revision Figure 1**). Adhibin treatment induced no significant changes in the expression levels, except for PXN, VASP, and KEAP1, which were slightly up- and down-regulated, respectively. The data suggest a compensatory response on specific cytoskeletal proteins and an inhibitory effect of adhibin on KEAP1, which has been described useful for the treatment of cancers with high NRF2 activity⁶. Recently it has been documented that brusatol-based inhibition of KEAP1/NRF2 in non-small-cell lung cancer cells resulted in the suppression of cell migration and invasion, with shrinking cell morphology due to decreased focal adhesions via inhibition of the RhoA-ROCK1 pathway⁷. Similar phenotypes are observed with adhibin treatment, which hints at a potential correlation of adhibin with KEAP1-regulation via the Rho/ROCK signaling cascade providing additional support to the proposed mechanism of adhibin interference in RhoA signaling. However, the above data are part of a broader analysis by our labs that is still ongoing, aiming at analyzing a broader spectrum of genes for correlation and expression profiling, further intended to be complemented by a large scale proteomic MS-based study.

With regard to the recommendation of using conventional chemotherapy drugs, we report in our manuscript experiments with the classical ROCK-inhibitor Y-27623, which targets the same signaling cascades as proposed for adhibin. The findings from these experiments support our hypothesis that the mechanism of action of adhibin relates to disturbances in the Myo9-controlled negative regulation of Rho-signaling. In future, we aim to use classical and new generation ROCK-inhibitors, such as fasudil, GSK269962, GSK429286, H1152Y27632, for which different potencies and specificities have been identified with promising preclinical and clinical efficacies as reported⁸.

2nd Revision Figure 1. Expression changes of key target genes in A549 cells. qPCR data showing relative changes in the expression of cytoskeleton-associated genes (A-B) and metastasis-related genes (C-D) upon adhibin treatment as heat map and bar diagram representation. The experiments were conducted as described in the qPCR section (Materials and Methods) in the main manuscript using appropriate primers. Ct values were calculated with the StepOne software version 2.1 with a cycle threshold of 0.2. Quantification of expression was performed with the $2^{-\Delta\Delta Ct}$ -method using the peptidylprolyl isomerase A (*PPIA*) as housekeeping gene. cDNA samples were analysed at least in triplicates.

References

1. Pacini, C. *et al.* A comprehensive clinically informed map of dependencies in cancer cells and framework for target prioritization. *Cancer Cell* **42**, 301-316.e9 (2024).
2. Birkbak, N. J. & McGranahan, N. Cancer Genome Evolutionary Trajectories in Metastasis. *Cancer Cell* **37**, 8–19 (2020).
3. Wang, H., Guo, M., Wei, H. & Chen, Y. Targeting p53 pathways: mechanisms, structures, and advances in therapy. *Signal Transduct Target Ther* **8**, 92 (2023).
4. Liu, J. *et al.* The regulation of PTEN: Novel insights into functions as cancer biomarkers and therapeutic targets. *J Cell Physiol* **238**, 1693–1715 (2023).
5. Śmiech, M., Leszczyński, P., Kono, H., Wardell, C. & Taniguchi, H. Emerging BRAF Mutations in Cancer Progression and Their Possible Effects on Transcriptional Networks. *Genes (Basel)* **11**, 1342 (2020).
6. Taguchi, K. & Yamamoto, M. The KEAP1–NRF2 System as a Molecular Target of Cancer Treatment. *Cancers (Basel)* **13**, 46 (2020).
7. Ko, E., Kim, D., Min, D. W., Kwon, S.-H. & Lee, J.-Y. Nrf2 regulates cell motility through RhoA–ROCK1 signalling in non-small-cell lung cancer cells. *Sci Rep* **11**, 1247 (2021).
8. Barcelo, J., Samain, R. & Sanz-Moreno, V. Preclinical to clinical utility of ROCK inhibitors in cancer. *Trends Cancer* **9**, 250–263 (2023).

Reviewer 2: The manuscript has been strengthened with the additional structural modelling and functional data, and the reviewer supports the publication of this work in Nature Communications.

- We thank the reviewer for the time and effort to review our manuscript and provide constructive comments and criticism.

Reviewer 3:

Q1. The authors have addressed most of my concerns. I do have a question/confusion regarding one of the experiments they have added in response to my questions. Specifically, the authors have performed an siRNA-mediated knockdown of Myo9B in B16-F1 and A549 cells to compare cells lacking Myo9B with the adhibin-treated cells (supp. Figure 5 and 7). The siRNA described in the Materials and Methods section are predesigned siRNAs from Qiagen that target human Myo9b. B16-F1 cells are mouse cells. Which siRNAs were used to target Myo9b in this cell line? In addition, Materials and methods provide information for siRNAs numbered 3, 4, and 5 while Supp. Figure 5 shows the results for siRNAs numbered 2, 3, and 5. I recommend that the authors carefully cross-check the siRNA information provided in the Materials and methods and the data.

- We thank the reviewer for the time and efforts, the constructive comments and thorough reading of our manuscripts. We apologize for the unintentional mistakes. The siRNAs used were #3, #4 and #5 as referenced in the Materials and Methods: siRNAs (#3, #4, #5) were obtained from QIAGEN with the reference numbers Hs_MYO9B_3 (SI00653709), Hs_MYO9B_4 (SI00653716), Hs_MYO9B_5 (SI03125661), and the scrambled RNA scRNA (4390847). The correction has been made to Suppl. Figure 5. These siRNAs were selected to target both, the human and the mouse MYO9B gene. We confirmed this by sequence analyses.

Q2. Minor comments: Supp. Figure 3 – the title reads “Adhibin impairs surface adherens“, should probably be “surface adhesion”

- We thank the reviewer. We corrected this.

Q3. Supp. Figure 4Q – quantification of the active Rho presents a decrease in 3 cell lines treated with 25 uM adhibin. However, the corresponding Western blot shown later in the supplemental data file shows treatment with 10um drug for the B16 cells.

- Indeed, the treatment for B16 cells was 10 μ M. We corrected this.

Answers to the reviewers

■ Answers to the reviewers:

Reviewer 1: Reviewer #1 (Remarks to the Author): I don't care about the data (Figure 1) provided by the author, I just care about the efficacy of the drug and its potential application prospects. So far, the author has refused to conduct critical research in this area at the animal level, thus unable to effectively evaluate the anti-tumor ability of drugs. I therefore do not agree to publish the paper under the current circumstances unless it can effectively evaluate the drug efficacy at the animal level.

- We have explained over the entire revision process, why additional experiments with animals cannot be addressed at the current stage. The permission for animal experiments has been granted by the Animal Welfare Service of Lower Saxony (German Civil Code BGB § 90) for experiments already included in the work, but not for anti-tumour models. Such animal experiments would be moving beyond the scope of a single paper.

Reviewer #2 (Remarks to the Author): No further comments.

Reviewer #3 (Remarks to the Author): All of my comments have been addressed. I recommend the manuscript for publication.

- We thank the reviewers for the time and efforts to review our manuscript.

Answer to the reviewer 1:

Reviewer 1: Reviewer #1 (Remarks to the Author): I don't care about the data (Figure 1) provided by the author, I just care about the efficacy of the drug and its potential application prospects. So far, the author has refused to conduct critical research in this area at the animal level, thus unable to effectively evaluate the anti-tumor ability of drugs. I therefore do not agree to publish the paper under the current circumstances unless it can effectively evaluate the drug efficacy at the animal level.

- In light of a missing permission to perform additional animal experiments to validate anti-tumor/anti-metastatic efficacy of the drug at the animal level further, we have highlighted in the manuscript the limitations of the study, particularly in the discussion. Firstly, we have specified in the abstract the exact cancerous cell models used, which comprise human and murine adenocarcinoma and melanoma cells as well as spheroids cultures. Secondly, to accentuate the non-pathological property of adhibin, we have provided data on serum drug distribution and show the histology of the organs. Thirdly, we point out the limitation of our study, referring to the necessity of in vivo cancer models to a) validate drug efficacy on the organismal level and b) assess risks and benefits for preclinical trials and potential therapeutic applications.”